# Transport Patterns of Global Aviation NO$_x$ and their Short-term O$_3$ Radiative Forcing – A Machine Learning Approach

Jin Maruhashi[1], Volker Grewe[1,2], Christine Frömming[2], Patrick Jöckel[2], Irene C. Dedoussi[1]

[1]Faculty of Aerospace Engineering, Section Aircraft Noise and Climate Effects, Delft University of Technology, Delft, the Netherlands
[2]Institut für Physik der Atmosphäre, Deutsches Zentrum für Luft- und Raumfahrt, Oberpfaffenhofen, Germany

*Correspondence to*: Irene C. Dedoussi (I.C.Dedoussi@tudelft.nl)

**Abstract.** Aviation produces a net climate warming contribution that comprises multiple forcing terms of mixed sign. Aircraft NO$_x$ emissions are associated with both warming and cooling terms, with the short-term increase in O$_3$ induced by NO$_x$ emissions being the dominant warming effect. The uncertainty associated with the magnitude of this climate forcer is amongst the highest out of all contributors from aviation and is owed to the non-linearity of the NO$_x$ – O$_3$ chemistry and the large dependency of the response on space and time, i.e., on the meteorological condition and background atmospheric composition. This study addresses how transport patterns of emitted NO$_x$ and their climate effects vary with respect to regions (North America, South America, Africa, Eurasia and Australasia) and seasons (January – March and July – September in 2014) by employing global-scale simulations. We quantify the climate effects from NO$_x$ emissions released at 250 hPa ($\sim$ 10 400 m) in terms of radiative forcing resulting from their induced short-term contributions to O$_3$. The emitted NO$_x$ is transported with Lagrangian air parcels within the ECHAM5/MESSy Atmospheric Chemistry (EMAC) model. To identify the main global transport patterns and associated climate impacts of the 14 000 simulated air parcel trajectories, the unsupervised QuickBundles clustering algorithm is adapted and applied. Results reveal a strong seasonal dependence of the contribution of NO$_x$ emissions to O$_3$. For most regions, a negative correlation is found between an air parcel's downward transport and its mean contribution to O$_3$. NO$_x$ emitted in the Northern regions (North America and Eurasia) experiences the longest residence times in the upper midlatitudes (40 – 45% of their lifetime), while those beginning in the South (South America, Africa and Australasia) remain mostly in the Tropics (45 – 50% of their lifetime). Due to elevated O$_3$ sensitivities, emissions in Australasia induce the highest overall radiative forcing, attaining values that are larger by factors of 2.7 and 1.2 relative to Eurasia during January and July, respectively. The location of the emissions does not necessarily correspond to the region that will be most affected – for instance, NO$_x$ over North America in July will induce the largest radiative forcing in Europe. Overall, this study highlights the spatially and temporally heterogeneous nature of the NO$_x$ – O$_3$ chemistry from a global perspective, which needs to be accounted for in efforts to minimize aviation's climate impact, given the sector's resilient growth.

# 1 Introduction

When anthropogenic influences are considered, the mean global surface temperature in 2011 – 2020 is estimated to have increased by 1.09 ℃ since pre-industrial times (1850 – 1900), according to the Sixth Assessment Report (IPCC, 2021). Studies have shown that aviation is accountable for approximately 3 – 5% of this total anthropogenic climate impact (Kärcher, 2018; Grewe et al., 2019; Lee et al., 2021). At the same time, the aeronautical industry has shown consistent growth in terms of revenue passenger kilometers (RPK) for many decades, even amidst global crises (Lee et al., 2009). As a result, aviation-

attributable greenhouse gas emissions have more than doubled between 1990 and 2017 (EEA, 2019). Despite the stagnation from air travel that was recently caused by the COVID-19 pandemic, the industry stresses the temporary nature of this disruption, indicating that aviation fuel burn and emissions will continue to grow in the coming decades (Boeing, 2020; Quadros et al., 2022a). The resilient growth of air traffic is ergo set to induce a growing climate impact in the upcoming years (Grewe et al., 2021).

Aircraft emissions affect climate via $CO_2$ and non-$CO_2$ effects. The latter include contributions from water vapor ($H_2O$) (Morris et al., 2003; Wilcox et al., 2012), nitrogen oxides or $NO_x$ ($NO+NO_2$) (Gauss et al., 2006; Gilmore et al., 2013; Köhler et al., 2013; Freeman, 2017), sulphur oxides ($SO_x$) (Kapadia et al., 2016), other aerosols like black carbon (BC) (Righi et al., 2013; Righi et al., 2016), and lastly via the formation of contrails (Schumann, 2005; Avila et al., 2019). This study focuses on nitrogen oxides, more specifically, on their short-term impact on ozone ($O_3$). $NO_x$ affects the climate by increasing the amount of ozone

and decreasing the levels of methane ($CH_4$), stratospheric water vapor (SWV) and the amount of background $O_3$ or Primary Mode Ozone (PMO) in the longer-term (Grewe et al., 2019), in addition to other pathways subjected to even larger uncertainties (e.g., nitrate aerosols) (Prashanth et al., 2022). The impact of these non-$CO_2$ effects depends on the aircraft's emission altitude (Matthes et al., 2021). For tropospheric $NO_x$ emissions, the production mechanism of $O_3$ is initiated via the hydroperoxyl radical ($HO_2$) and nitric oxide (NO), followed by the photodissociation of nitrogen dioxide ($NO_2$) along with the combination

of atomic (O) and molecular ($O_2$) oxygen atoms. Within the upper troposphere/lower stratosphere (UT/LS), $NO_x$ would still lead to a quasi-linear production of $O_3$ (Matthes et al., 2022). Slightly above the UT/LS, between 13 – 14 km in altitude, one finds the $O_3$-neutral region in which emitted $NO_x$ would lead to a net null $O_3$ disturbance seeing as its tropospheric production is counteracted by its stratospheric destruction. Above this region, $O_3$ is consumed by NO to produce $NO_2$ and $O_2$ (Fritz et al., 2022). The ability of nitrogen oxides to oxidate methane, producing water vapor in the process, results in a reduction in methane

concentration, this eventually leads to a decrease in the production of stratospheric water vapor as less $CH_4$ enters the stratosphere (Myhre et al., 2007). Combining these effects yields a cooling effect, as both are greenhouse gases. In the long run, a decrease in ozone concentration is also induced, since $CH_4$ is a precursor to $O_3$ (Brasseur, 1998).

When all of these aviation $NO_x$ effects are compared on a common scale for the year 2018, the short-term increase in $O_3$ is the only contributor to a climate warming effect, in terms of the effective radiative forcing (ERF) metric (IPCC, 2013), and is

estimated to produce a net warming of 17.5 mWm$^{-2}$ [0.6-29, 90% confidence interval (CI)]. It is also known that the largest

radiative forcing (RF) uncertainty arises from this short-term increase (e.g. Grewe et al., 2019). Short-term $NO_x$ effects also induce the second highest ERF across aviation's climate forcers with an estimate of 49.3 mWm$^{-2}$ [32-76, 90% CI] (Lee et al., 2021).

The climate impact associated with aviation $NO_x$ emissions has been studied in terms of varying the cruise flight altitude
(Frömming et al., 2012; Matthes et al., 2021), emission locations (Stevenson and Derwent, 2009; Köhler et al., 2013), seasons (Gilmore et al., 2013; Stevenson et al., 2004), background $O_3$ concentrations (Dahlmann et al., 2011) and weather patterns (Frömming et al., 2021; Rosanka et al., 2020). In regard to research focusing on the spatial relationship between emission location and $NO_x$ environmental effects, the work by Stevenson and Derwent (2009) may be referred. Therein, an $O_3$ anomaly is introduced and followed via the STOCHEM Lagrangian Chemistry Transport Model (CTM) across 9 distinct regions, where
it was concluded that emissions in less polluted areas (lower $NO_x$ levels) led to larger instantaneous radiative forcings (IRF) compared to those in which the emission occurred at locations of larger background $NO_x$ concentrations. In other words, flying over cleaner areas is more likely to generate more substantial climate impacts. Skowron et al. (2015) found a significant geographic dependence, through a sub-division of the globe in 10 regions, between emissions and the subsequent radiative forcing according to simulations from the MOZART-3 model. They found that the largest $O_3$ burden and consequent
disturbance in RF occurred in Australia and the lowest in Europe, due to varying levels of ozone production efficiencies. Overall, the North Atlantic and tropical regions in Brazil showed the strongest net $NO_x$ RF effects. Köhler et al. (2013) examined aircraft $NO_x$ in four regions (USA, Europe, India and China) with the p-TOMCAT chemistry transport model (CTM) and found that lower latitude emission perturbations have RF magnitudes that are 6 times larger than those due to emissions at higher latitudes, agreeing well with findings from Grewe and Stenke (2008). Gilmore et al. (2013) also found larger ozone
sensitivities in the Tropics, particularly near the Australasian region. Recent studies by Frömming et al. (2021) and Rosanka et al. (2020) have adopted a Lagrangian approach to investigate the impact of emitting $NO_x$ and $H_2O$ in different weather conditions across Summer and Winter in the North Atlantic using the EMAC model. The former investigated trajectories starting at emission regions within and to the west of a high-pressure ridge, analyzing how they contribute differently to $O_3$. In summary, none of these examples have comprehensively studied the complete and intercontinental journey of aviation $NO_x$
from its point of release, intermediate transport pathways, until most of the resulting $O_3$ is removed from the atmosphere 3 months later (typical $O_3$ lifetime), and the associated climate forcing. The present study therefore seeks to bridge this gap by characterizing this multi-stage, spatio-temporal evolution of emissions on a large scale across several regions globally during different seasons.

A Lagrangian approach rests on the idea of following an individual air parcel in space as a function of time. The evolution of specific quantities, such as mixing ratios of chemical species, may be tracked along an air parcel's pathway. The method thus allows for a complete and detailed accompaniment of the different trajectories that gas-phase emissions may take on an individual basis. Such an approach, however, naturally yields large amounts of data (in the order of terabytes) from the

necessity to include close to 170 000 air parcels per simulation to both ensure realistic transport and inter-parcel mixing. All of these trajectories can, however, be efficiently grouped together with the integration of unsupervised machine learning techniques such as clustering. In other words, clustering offers a systematic way of categorizing thousands of air parcels based on common features, which ultimately allows for a faster identification of patterns in very large datasets. Outside of aviation and atmospheric sciences, trajectory clustering has been applied in various contexts to improve pattern recognition of e.g., deer movements, cyclist trails, pedestrian paths and naval vessel traffic (Melssen, 2020; Lee et al., 2007). Within aviation, aircraft routes have been grouped according to methods like the Density-Based Spatial Clustering of Applications with Noise (DBSCAN) or the Hierarchical DBSCAN (HDBSCAN) as applied by Corrado et al. (2020), Basora et al. (2017), Olive and Basora (2019) and Olive and Morio (2018). Within atmospheric sciences, other techniques like K-means clustering have been used to systematically categorize different types of aerosol regimes at distinct altitudes and regions (Li et al., 2022). Another example is Kassomenos et al. (2010) who applied several clustering approaches like K-means and Self-Organizing Maps (SOM) to isolate the pathways with highest $PM_{10}$ concentrations. Clustering has also been applied to output from the ECHAM5/MESSy1 model and $O_3$ observations by Tarasova et al. (2007), in which an agglomerative hierarchical clustering algorithm with a squared Euclidean distance metric was used to discern patterns in the distribution of surface $O_3$ mixing ratios in the extra-tropics. In our study, trajectory clustering is performed with an algorithm from neuroscience called QuickBundles with a newly proposed clustering function. To the authors' best knowledge, this is the first time that a global characterization of Lagrangian air parcel trajectories from the EMAC model is performed with this approach.

We perform Lagrangian simulations with the EMAC model in which aviation $NO_x$ emissions are released in 5 regions (North (N.) America, South (S.) America, Africa, Eurasia and Australasia) at a representative aircraft cruise level altitude of 250 hPa in January and July of 2014 and traced in terms of their chemical impact (via their $O_3$ atmospheric disturbance for 3 months following their release). Overall, this study addresses the following research questions: (1) What are the main aviation $NO_x$ transport pathways depending on when and where they are emitted, and which ones exhibit the largest $O_3$ production? (2) Under which conditions are the radiative forcings largest and where are these effects expected to occur?

The current study is divided into 5 sections. Section 2 describes the methodology and defines the main features and settings of the EMAC chemistry-climate model used. It further provides details regarding the QuickBundles clustering algorithm. In the third section, the main results for clustering, $O_3$ contributions and RF are shown, both in terms of which locations yield the highest mean global impact and which regions are most affected by an emission in a given region. This is then followed by a discussion of the findings in Section 4. Section 5 delivers a summary of key points along with concluding remarks with suggestions for future research.

## 2 Methodology

This section describes the EMAC model setup and relevant simulations that were performed (Section 2.1). For every region (as described in Section 2.2), 28 emission points were defined from which the emitted $NO_x$ is divided onto 50 air parcels. A total of 28 median trajectories is then derived for each point to summarize the transport behavior of the aforementioned 50 air parcels for each of the 5 regions (N. America, S. America, Eurasia, Africa and Australasia). Based on these emission regions, a source-receptor analysis is performed, for which the receptor boundaries are defined in Section 2.3. The altitude and latitude along with the radiative forcing for these median trajectories then serve as input for the QuickBundles clustering algorithm (Section 2.4) and the choice for this particular clustering method from among other viable alternatives is justified. Lastly, in Section 2.5, the definition of the rate of descent is introduced.

### 2.1 The EMAC model and relevant sub-models

The numerical chemistry simulations in this study were performed with the European Centre for Medium-Range Weather Forecasts – Hamburg (ECHAM)/Modular Earth Sub-model System (MESSy) Atmospheric Chemistry (EMAC) model. It is a climate-chemistry model that allows for the base General Circulation Model (GCM) (here ECHAM5, Roeckner et al., 2003), to be coupled to other sub-models via the more recent second version of the Modular Earth Sub-model System (MESSy2). These sub-models represent specific atmospheric and chemical processes interacting with the biosphere, land and ocean throughout the troposphere and middle atmosphere (Jöckel et al. 2010). In this study, a T42L41 spectral resolution is chosen to provide a balance between computational costs and resolution. The L-value indicates that 41 discrete vertical hybrid pressure levels ranging from the surface up to the uppermost layer centered at 5 hPa are being used while the T-value defines the triangular spectral truncation that corresponds to a horizontal discretization of the grid space into 128 longitude and 64 latitude points, equivalent to a 2.8º × 2.8º quadratic Gaussian grid. A total of 10 simulations (5 regions × 2 seasons) were performed, consuming approximately 35 000 CPU hours of computation time on the Dutch supercomputer Cartesius (now Snellius). The first season encompasses a 3-month period spanning January 1 – March 31, while the second season ranges from July 1 – September 30, both in 2014. Variables including the vorticity, the logarithm of the pressure field at the surface, the divergence of the wind and the temperature within these simulations were nudged by Newtonian relaxation towards 2014 ERA-Interim reanalysis data.

Version 5.3.02 of the ECHAM5 base model and MESSy version 2.55.0 were used. Three sub-models are of particular relevance to this study: TREXP (Tracer Release EXPeriments from Point Sources; Jöckel et al., 2010), ATTILA (Atmospheric Tracer Transport In a LAgrangian model; Brinkop and Jöckel, 2019) and AIRTRAC (see supplement from Grewe et al., 2014). The first is used to define the positions of the $NO_x$ pulse emissions in terms of their latitude, longitude and pressure altitude in each of the 5 regions (see Fig. 1). A total of 28 emission points per region (see Table A1 in Appendix A) at an altitude of 250 hPa at 06h00 UTC was chosen to be representative of an aircraft flying at a typical cruise level in a given region. This emission time choice brings forth the added benefit of making our results more comparable to previous research (Grewe et al., 2014;

Rosanka et al., 2020; Frömming et al., 2021). In each emission point, an amount of $5\times10^5$ kg (NO) is injected into the atmosphere within a 15-minute time step (the output frequency for the air parcel spatial coordinates is 6 hours) and 50 air parcel trajectories are then pseudo-randomly initialized according to a uniform distribution between 0 and 1 within the grid cell of each emission point. This amount of emitted NO may be compared to total yearly NO emissions at cruise (~250 hPa) by commercial aircraft for all five regions (defined in Fig. 1) from the most recently available aviation emissions inventory

(Quadros et al., 2022b). According to this inventory, the emitted NO amount of 0.5 Gg constitutes roughly 40% of mean total yearly emissions by commercial aircraft in N. America, 32% for Eurasia, 186% for S. America, 323% for Africa and 118% for Australasia. For the linearized AIRTRAC sub-model itself, however, the amount of NO emitted is not of particular relevance since the resulting volume mixing ratios will be scaled linearly according to changes in this emission quantity. In terms of the background $NO_x$ levels, typical volume mixing ratios near N. America in July 2014 at 250 hPa range between

$5\times10^{-10}$ and $9\times10^{-10}$ mol·mol$^{-1}$. With this Lagrangian approach, a single simulation is sufficient to calculate the contribution of a $NO_x$ emission to atmospheric $O_3$ for all 28 emission points in parallel for a given region based on the 50 emission-carrying trajectories that are released at each of these emission points (Grewe et al., 2014). The optimal choice for this number of parcels derives from a sensitivity study performed by Grewe et al. (2014) in which the standard deviations of $NO_x$ and $O_3$ mixing ratios were found to already decrease to under 10% if more than 20 trajectories are used. This therefore equates to having 50

emission-carrying air parcels (for a total of $28\times50\times5 = 7\,000$ globally) that are initially responsible for the transport of $NO_x$ to the rest of the world. There are approximately 170 000 other trajectories that are initialized outside of the grid cells with emissions, also following a uniform distribution from 0 to 1, so as to simulate diffusive effects and mixing between air parcels. These air parcels are advected by the ATTILA sub-model according to the 3-dimensional wind field of the base model ECHAM5. The mixing itself between Lagrangian parcels depends on dimensionless mixing coefficients defined per vertical

layer (e.g., the troposphere) and is parameterized via the LGTMIX (LaGrangian Tracer MIXing) sub-model (Brinkop and Jöckel, 2019). Along each of these air parcel trajectory points, the AIRTRAC sub-model then calculates the contribution of the $NO_x$ emissions to the atmospheric composition of $O_3$, $CH_4$, $HNO_3$, OH and active nitrogen species ($NO_y$) by also considering the background concentrations computed by the sub-model MECCA (Module Efficiently Calculating the Chemistry of the Atmosphere, Sander et al., 2011) for the troposphere and stratosphere, as is described by Frömming et al.

(2021) and Rosanka et al. (2020). The effect of a $NO_x$ disturbance on the mixing ratios of other chemical species like $O_3$ can be understood in terms of an interaction between production and loss terms in which $O_3$ production is initiated by the reaction $NO + HO_2 \rightarrow OH + NO_2$, followed by the photolysis of $NO_2$ ($NO_2 + hv \rightarrow NO + O$) and finally the combination of oxygen and dioxygen ($O + O_2 \rightarrow O_3$). The calculation of these photolysis rate coefficients is performed by the JVAL module (Sander et al., 2014). $O_3$ loss is simulated in terms of the reaction $NO_2 + O_3 \rightarrow NO + 2O_2$, with the main loss mechanism being the

reaction of $O_3$ with hydrogen oxides ($HO_x$), i.e., $OH + O_3 \rightarrow HO_2 + O_2$ and $HO_2 + O_3 \rightarrow OH + 2O_2$. The net $O_3$ calculated by AIRTRAC (see Fig. B1 in Appendix B) results from $O_3$ production and loss terms (ProdO3N and LossO3N, respectively) and an $O_3$ loss term from non-Nitrogen species (LossO3Y). Scavenging processes and dry deposition are also contemplated in

this sub-model. Further technical details regarding this simulation setup have been documented by Frömming et al. (2021) and Grewe et al. (2014).

The radiative forcing from $NO_x$-induced $O_3$ is calculated using the RAD sub-model (Dietmüller et al., 2016) in which the IRF for all available vertical levels, given the selected resolution, is computed after transforming the Lagrangian $O_3$ disturbance field into grid-point space. By radiative forcing we are not referring to the more common definition that pertains to the difference in RF contributions since pre-industrial times up to a given year, rather, we define it as the mean radiative impact arising from a pulse emission. The radiative forcing relative to the climatological tropopause is the chosen measure in this

study, since the stratospheric-adjusted RF and the ERF are not applicable to the pulse emissions used. Warming occurs whenever IRF > 0 and cooling when IRF < 0. The net change in RF for $O_3$ is calculated as the difference of the forcings between two ozone fields: 1) the sum of the background $O_3$ and the additional $O_3$ disturbance from the $NO_x$ emission and 2) the background $O_3$ volume mixing ratio (undisturbed scenario). Given the current simulation setup, the RAD sub-model can calculate the long- and shortwave fluxes for the $O_3$ disturbance field arising from the emitted $NO_x$ that is transported by 50 air

parcels at each emission point in Fig. 1. It is therefore not possible to track the individual $O_3$ disturbance fields generated by each of these 50 air parcel trajectories. A one-to-one association between radiative flux and geometry is then established via the median trajectory, which more realistically represents the dynamical characteristics of a set of trajectories when compared to the mean. In our setup, the RAD sub-model computes short- and longwave fluxes every 2 hours and we use the values that are computed instantaneously at every 6-hour output frequency.

**2.2 $NO_x$ emission locations**

Each "+" in Fig. 1 represents the point at which a $NO_x$ emission was released as well as the location in which 50 Lagrangian trajectories were pseudo-randomly initialized within the grid cell that comprises this emission. A total of 28 emission points per region were used as a compromise between the level of detail in the analysis for all five regions and the available computational resources. While these emission points are not representative of the present-day spatial distribution of aviation

emissions (e.g. elevated flight traffic like the North Atlantic flight corridor), the aim of this work is to understand the underlying processes at a global scale, and as such we distribute the emission points at a wider spatial scale globally. In addition, the predictions for the shift in global flight distribution indicate a likelihood of reduced traffic in the North Atlantic given likely decreases in RPK from 26% to 17% and from 23% to 17% for North America and Eurasia respectively (Gössling and Humpe, 2020). As a result, knowledge on less commonly flown areas at the present time will also be needed to accompany studies that

already provide an in-depth examination of aviation climate effects for current flight patterns (Grewe et al., 2014; Frömming et al., 2021; Rosanka et al., 2020; Grobler et al., 2019).

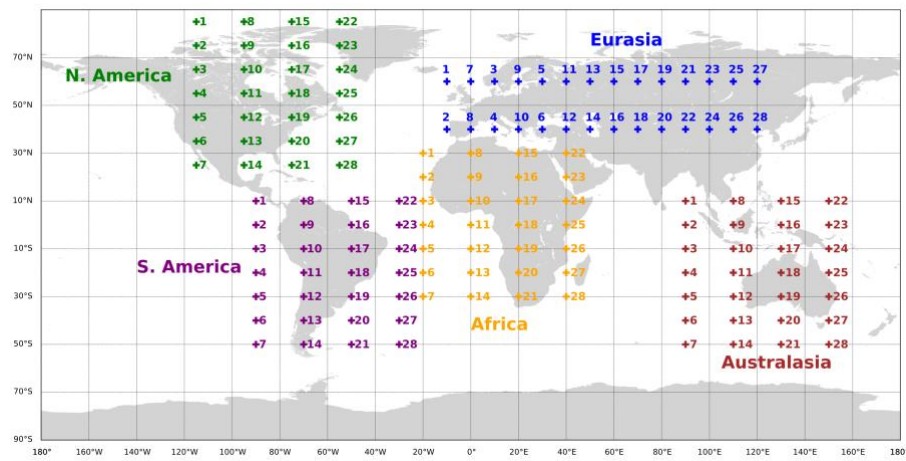

**Figure 1 –** Numbered NO$_x$ point source emissions by region. Emission point coordinates are included in Appendix A.

### 2.3 Source-receptor analysis

To better understand the relation between the location of emission and the location of largest radiative forcing, and to therefore
answer the second research question, a source-receptor analysis is conducted. Past studies, within the context of aviation air
quality effects, have adopted a similar approach to comprehend if, in terms of aviation-induced O$_3$ and fine particulate matter
(PM$_{2.5}$), the source of the emission directly corresponded to the most affected region (Quadros et al., 2020). The regional
division in Fig. 2 is inspired by Stevenson and Derwent (2009), who similarly analyzed which areas would exhibit the largest
instantaneous RF based on an aviation NO$_x$ emission in N. America. Here, we look at aviation climate effects by considering
the instantaneous RF that is calculated with the RAD sub-model with respect to the climatological tropopause and subsequently
averaging it over the receptor regions in Fig. 2.

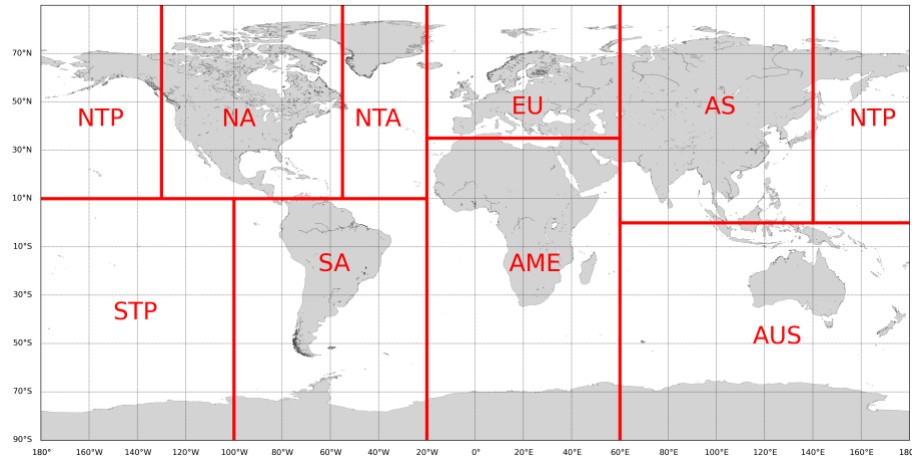

**Figure 2 –** Regional divisions for source-receptor analysis based on RF (NTA: Northern Trans – Atlantic, NA: North America, NTP: Northern Trans-Pacific, EU: Europe, AS: Asia, STP: Southern Trans-Pacific, SA: South America, AME: Africa & Middle East and AUS: Australia).

## 2.4 Trajectory clustering

To systematically group the output trajectories of the Lagrangian approach within EMAC (Section 2.1) in terms of their geometric similarities in altitude and latitude, as well as their radiative forcing effects, we incorporate a clustering algorithm to the methodology. This enables us to first identify the different types of transport geometries across all regions and then to associate an average radiative forcing estimate to each in an orderly fashion. First, for each emission point in each region, a representative (median) trajectory is computed (see red lines in Fig. 5). For each region, these median trajectories are clustered using the QuickBundles approach, as described in Section 2.4.1. The air parcel trajectories that are clustered in this study are therefore all median trajectories, each being constructed from the median values of longitude, latitude and altitude. A total of 28 median trajectories are generated per region, each is defined from the 50 trajectories that are initialized within the grid cell comprising the $NO_x$ emission (see Section 2.2).

## 2.4.1 QuickBundles Application to Median Air Parcel Trajectories

QuickBundles (Garyfallidis et al., 2012) was originally developed to help neuroscientists group large datasets arising from magnetic resonance imaging, called tractographies (consisting of streamlines in the order of $10^6$ that represent cerebral nerve tracts), and therefore simplify the subsequent analyses. Here, we apply this method to air parcel trajectories. We have selected this method for three reasons: QuickBundles (1) was designed to cluster nerves in the form of streamlines, which greatly resemble air parcel trajectories, (2) facilitates the integration of a customizable distance metric or clustering function (used for the attribution of the trajectories into clusters), and (3) has a low computational cost compared to alternative methods because it is, on average, linear in complexity, i.e., $O(n)$ (Garyfallidis et al., 2012). The QuickBundles tractography clustering software has been implemented in Python 3 and is openly available at dipy.org. Figure 3 provides an overview of the QuickBundles algorithm as a sequence of five steps, when it is applied to median air parcel trajectories. The median trajectory is chosen as it is composed entirely of original coordinate points, unlike the more artificial mean trajectory. This allows for the median trajectory to more accurately represent the dynamical behavior of a set of trajectories, as is seen in Fig. 5 for emission points 24 and 28 for N. America (Figs. 5 (a) and (b) respectively).

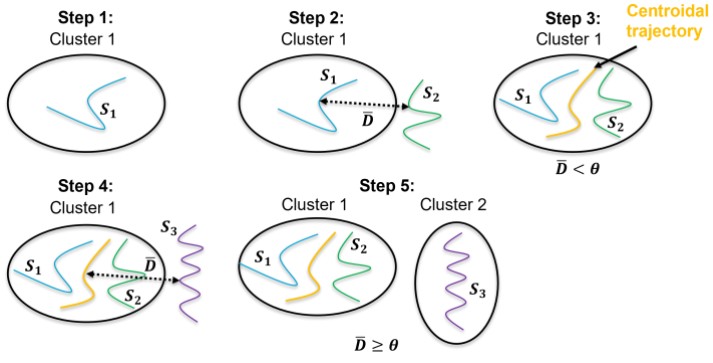

**Figure 3** – Illustration of the five steps of the QuickBundles clustering algorithm. Air parcel trajectories are denoted by $S_1$ through $S_3$, $\theta$ is the user-defined clustering threshold and $\overline{D}$ is the mean elementwise distance between trajectories according to Eq. 1 (for details see text).

In the first step, air parcel trajectory $s_1$ is directly placed within the first cluster as there are no pre-existing groups. At this stage, the centroidal trajectory, which is the average path of all trajectories within a given cluster, coincides with the coordinates of $s_1$. In step 2, the mean elementwise distance $\overline{D}$ between the centroidal trajectory of cluster 1 and the next trajectory $s_2$ is calculated. The method for computing this value varies according to the distance function that is defined by the user. One of the simplest and most widely used metrics is the Euclidean distance in which a 3-dimensional length is calculated based on the latitude, longitude and altitudes (Corrado et al., 2020; Olive and Basora, 2020). However, this metric has been adapted in our study to account for the impact of $NO_x$ emissions on the Earth's energy balance during the clustering process by replacing the longitude of the air parcels with the radiative forcing, where the former variable exhibits less variation between the trajectories in each region than the latter. This dimensionless, averaged function $\overline{D}(s_i, s_j)$ is defined by Eq. (1) with $\phi$ being the latitude in degrees, $H$ the altitude in km and $\overline{RF}$ the mean radiative forcing per emission point in W·m⁻²:

$$\overline{D}(s_i, s_j) = \frac{1}{T} \sum_{t=1}^{T} \sqrt{\left(\frac{\phi_t^i - \phi_t^j}{\max \phi}\right)^2 + \left(\frac{H_t^i - H_t^j}{\max H}\right)^2 + \left(\frac{\overline{RF_t^i} - \overline{RF_t^j}}{\max \overline{RF}}\right)^2}. \tag{1}$$

Given that the range of latitudes varies from $-90°$ to $90°$ and that the altitude is between the surface and the point of release at 250 hPa, the maximum values are $\max \phi = 180°$ and $\max H \approx 10$ km. This newly adapted and normalized Euclidean average distance function proposed here describes similarities in both the geometry and radiative forcing of air parcels, and is performed between a pair of 3-dimensional trajectories $s_i$ and $s_j$ with the following structures: $s_i = [(\phi_1^i, H_1^i, \overline{RF_1^i}), (\phi_2^i, H_2^i, \overline{RF_2^i}), \dots, (\phi_T^i, H_T^i, \overline{RF_T^i})]$ and $s_j = [(\phi_1^j, H_1^j, \overline{RF_1^j}), (\phi_2^j, H_2^j, \overline{RF_2^j}), \dots, (\phi_T^j, H_T^j, \overline{RF_T^j})]$ for which $T$ represents the total number of discrete time steps along a trajectory. The mean radiative forcing estimate per emission point from RAD is calculated for each of these 28 points per region and this value then becomes the third component in each triple $(\phi_1^i, H_1^i, \overline{RF_1^i})$ for $s_i$ and $s_j$. All air parcel trajectories are discretized with the same time step length of $\Delta t = 15$ minutes and are followed for 3 months after the emission date. The value of $T$, however, is 360 given that the model output frequency for the air parcel spatial coordinates is 6 hours (3×30×24/6). The normalizations of Eq. (1) are necessary, given the largely differing magnitudes of $\phi$, $H$ and $\overline{RF}$. Once the average distance $\overline{D}$ is found, the result is compared to a user-defined clustering threshold $\theta$. If $\overline{D} < \theta$, as in step 3 of Fig. 3, the candidate trajectory (green) is added to the first cluster and the coordinates of the centroidal trajectory (yellow) are updated. This on-the-fly updating of the centroidal trajectory avoids the need for a full recalculation within the cluster itself, thereby speeding up the overall algorithm (Garyfallidis et al., 2012). If, on the other hand, $\overline{D} \geq \theta$ after comparing $s_3$ with the centroidal trajectory (step 4), then a new cluster is created for this third candidate trajectory $s_3$ (step 5). The process is continued for the remaining trajectories. This approach to process Lagrangian air parcel

trajectories from the ATTILA sub-model using the distance metric (=clustering function) of Eq. (1) is novel. QuickBundles has, however, been applied in atmospheric sciences to cluster cloud movements from radar imagery (Edwards et al., 2019).

In summary, the QuickBundles algorithm systematically groups median trajectories based on geometric similarities (altitude and latitude) and their climate effects (mean radiative forcing associated with an emission point). The first trajectory simply becomes the first cluster and all subsequent trajectories are allocated based on the calculation of a metric (Eq. 1) that we propose here that accounts for both transport features and climate effects. At every step in the clustering process, the distance between the centroidal trajectory of a cluster and the candidate trajectory is calculated using Eq. 1 and depending on the user-

defined parameter $\theta$, it will either be placed within a pre-existing cluster or in an entirely new one. The optimal selection of $\theta$ is discussed in the next section.

### 2.4.2 Selection of an optimal threshold $\theta$

One of the challenges of applying clustering techniques is determining the optimal number of clusters, which in the case of QuickBundles, is determined via the user-defined threshold $\theta$. It is typical to combine the evolution of a sum of squared errors

(SSE) function with silhouette coefficients to arrive at the most efficient number of clusters, as has been done in a recent aerosol clustering study with a K-means method (Li et al., 2022). This is a grouping technique that seeks to distribute data points across an optimally chosen number of clusters based on the minimization of a within-cluster sum of squares (Hartigan and Wong, 1979). The approach chosen here is based on the L-method, developed by Salvador and Chan (2005) and applied by Kassomenos et al. (2010). It consists of finding the intersection between two linear regressions that model each half of a

curve generated from plotting the evolution of a trajectory-based metric, specifically here the average distance within clusters, as a function of the cluster number, as is exemplified in Fig. 4 (a). Possible other metrics as described by Kassomenos et al. (2010) could be the SSE, a similarity function, or another distance metric. The red dot in Fig. 4 (a) may be viewed as a partitioning point as it divides the curve in two quasi-linear sections: left and right. The final linear regression (dotted red line) for the points on the left may be obtained iteratively by beginning with the most basic case of regressing the first two, leftmost

data points. The fit of such a linear function may then be improved gradually by adding more and more points in each iteration. According to Salvador and Chan (2005), the optimal distribution of points to be regressed by each side could be achieved by iteratively selecting the partitioning point that minimizes the total root mean squared error of both linear regressions. Kassomenos et al. (2010), whose approach we follow here, however, have also tested this method by applying different curve modelling techniques that include higher order polynomials and splines and obtained consistent results relative to the linear

approach.

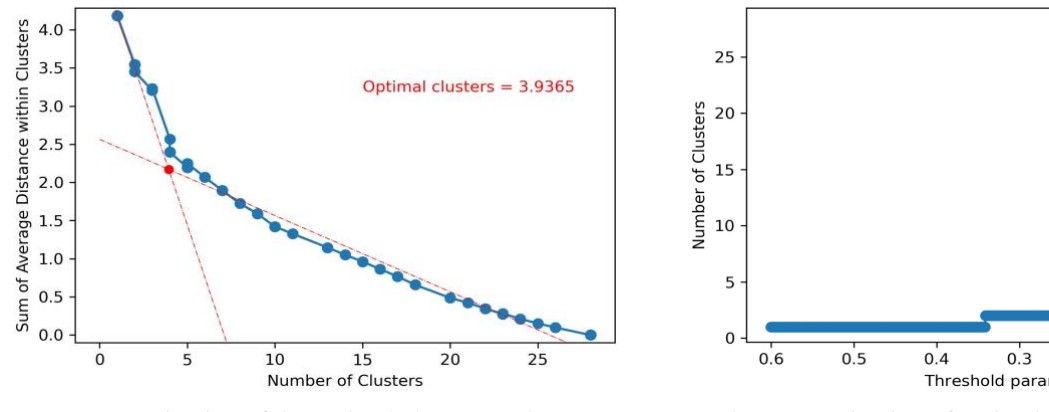

(a) Determination of the optimal cluster number      (b) Determination of optimal $\theta$ from cluster number

**Figure 4** – Determination of optimal clustering threshold $\theta$ for N. America during July – September 2014. (a) Sum of averaged distance $E(\theta)$ as a function of the number of clusters (blue). Two linear regressions (red) are added. The intersection (red dot) is giving the optimal number of clusters. (b) Variation of the number of clusters with $\theta$.

The evaluation metric $E(\theta)$, given by Eq. (2), is the sum of mean "distances" (based on the adapted Euclidean function of Eq. 315 (1)) to the centroidal trajectory within a cluster $c_k$ with $k = 1,2,\dots,N(\theta)$, where $N$ is the total number of clusters that depends on $\theta$, across all clusters. The number of median trajectories within a cluster $c_k$ is given by $L(c_k)$ (and therefore also depends on $\theta$), and $\bar{\mu}_k$ is the centroidal trajectory for the $k^{th}$ cluster. Radiative forcing is included in the clustering process via Eq. (1) and therefore also in the evaluation metric in Eq. (2). This evaluation metric, as well as the L-method, are applied entirely to median trajectories, which are shown in red in Fig. 5.

$$E(\theta) = \sum_{k=1}^{N(\theta)} \sum_{l=1}^{L(c_k)} \overline{D}(s_l, \bar{\mu}_k) \tag{2}$$

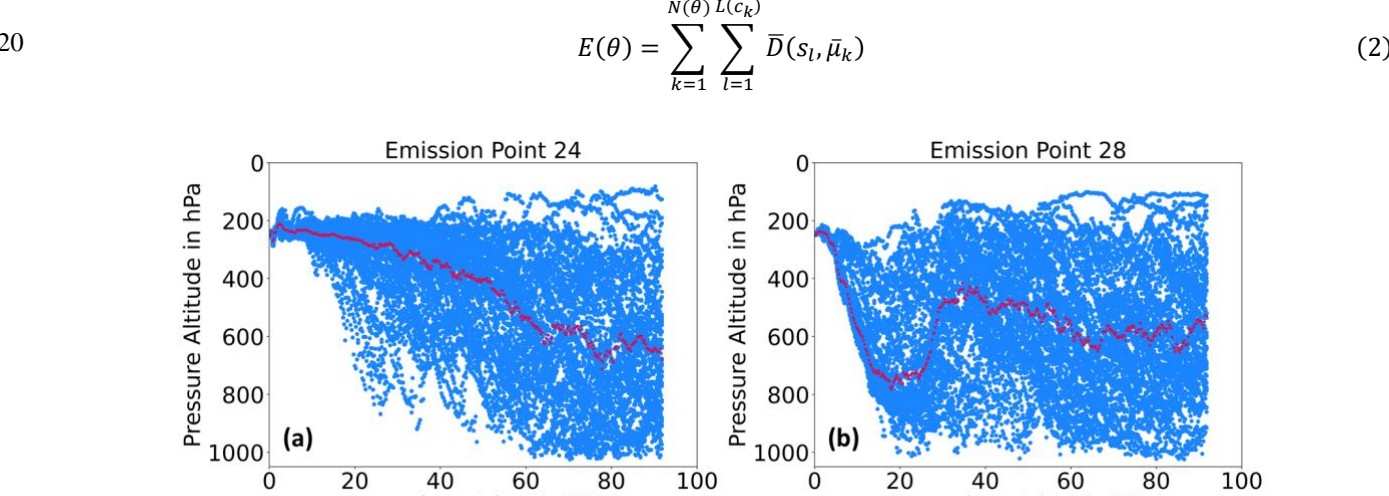

**Figure 5** – Temporal evolution of the air parcel trajectory pressure altitude of median trajectories (red) based on 50 Lagrangian paths (blue) in (a) emission point 24 and in (b) emission point 28 for N. America for July – September 2014. The trajectories

of the remaining regions are included in the Supplementary Figures. Figure S7 shows all 28 emission points for N. America during July – September 2014.

Since QuickBundles uses $\theta$ as the user input, a range of values of $\theta$ is tested with Eq. (2) until an optimal cluster number is found according to Kassomenos et al. (2010) via the intersection of two regression lines (red dotted lines in Fig. 4 (a)). Once the optimal cluster number is found, the corresponding value of $\theta$ is chosen (Fig. 4 (b)). The calculations of the optimal cluster numbers for all regions and seasons are part of the supplementary material. For N. America, for instance, Fig. S3 and S9 illustrate this process for January and July respectively.

### 2.5 The rate of descent $\dot{H}$


To quantitatively distinguish between vertical transport patterns, we use the rate of descent $\dot{H}$ in hPa·day$^{-1}$, defined here according to Eq. (3). It is the rate of change in altitude of an air parcel from its point of release until it first reaches its global minimum altitude at a time $t^*$. As the release altitude is constant in all scenarios, $H(0) = 250$ hPa. The quantity $\dot{H}$ is calculated for all regions in Section 3.1 and is intended as a metric to understand if the initial downward transport has an impact

on the atmospheric concentrations of $O_3$.

$$\dot{H} = \frac{|H(0) - H_{min}(t^*)|}{t^*} \tag{3}$$

### 3 Results

We first present how the $NO_x$ transport patterns vary depending on the location and season of emission by relating the rate of air parcel descent with the resulting $O_3$ production and estimating the latitudinal residence times of the 1 400 Lagrangian

trajectories in each region (Section 3.1). The median trajectories are then grouped into clusters and assessed in terms of their characteristic transport pathways (Section 3.2). Finally, the spatial characteristics of the resulting RF from $NO_x$ over different regions is presented (Section 3.3).

### 3.1. Seasonality and meteorological conditions as drivers of $NO_x$ transport and chemistry

We use the air parcel rate of descent (as defined in Eq. 3) to describe the differences in the resulting $O_3$ disturbances induced

by different $NO_x$-carrying air parcels. Since the $O_3$ lifetime varies by altitude, a lower rate of descent $\dot{H}$ (and thus greater residence time at a higher altitude) is expected to likely result in a higher $O_3$ disturbance than an air parcel with a faster downward transport (larger $\dot{H}$). Figure 6 depicts the variation of the mean $O_3$ disturbance with $\dot{H}$ of the median trajectories for all regions during both of the seasons. Apart from Africa during July, between 50-100% of the top 20% maximum $O_3$ disturbances show $\dot{H} < 10$ hPa·day$^{-1}$, indicating that longer residence times near the altitude of emission lead to a larger impact

on the atmospheric composition of $O_3$. This inverse relationship between $\dot{H}$ and mean $O_3$ is likely owed to faster removal rates for $O_3$ at lower altitudes compared to more efficient accumulation of chemical species at higher altitudes, therefore leading to

lower chemical lifetimes for $O_3$ in the lower troposphere (Fig. 4 in Grewe et al., 2002a; Frömming et al., 2012; Gauss et al., 2006). We note that the rate of descent for a 40-day window, as opposed to the 90-day window presented here, performs slightly better in capturing the immediate downward behavior of certain transport patterns, as is the case for emission point 19 in Eurasia in July (see supplementary figures, Fig. S25), but the overall negative trend relative to the mean $O_3$ disturbance is maintained. We note that the same negative correlation is found, if the mean $O_3$ disturbance is replaced with the radiative forcing that it induces.

To ascertain whether a statistically significant correlation exists between the average descent rates $\dot{H}$ and the mean $O_3$ contribution, as well as to quantify the strength of this correlation, three metrics are calculated: the Pearson, Kendall and Spearman rank correlation coefficients. These have also been applied by Rosanka et al. (2020) to better understand the interactions between relevant chemical species, i.e., $NO_x$ and $O_3$ volume mixing ratios. Table 1 summarizes the values for these coefficients for all regions for emitted $NO_x$ during January and July. During January, the correlation coefficients are all negative and, with the exception of Eurasia, are also statistically significant based on a level of significance of $\alpha = 5\%$, since their p-values are below this threshold. The strongest negative correlations exist for S. America and Africa, where all 3 metrics are statistically significant with a magnitude of about 0.5. During both periods for air parcels released in Eurasia, no significant correlation is found between $\dot{H}$ and the mean $O_3$ contribution. The only statistically significant positive correlation is found for Africa during July, according to the Spearman and Kendall coefficients. Australasia during July appears to not exhibit any correlation at all.

**Table 1** – Statistical correlations between $\dot{H}$ and mean $O_3$ contribution using the Pearson, Kendall and Spearman correlation coefficients. P-values are shown for each corresponding metric. The null hypothesis assumes that no correlation exists between $\dot{H}$ and the mean $O_3$ contribution.

| | January | | | | | | July | | | | | |
|---|---|---|---|---|---|---|---|---|---|---|---|---|
| | Pearson | p-value | Spearman | p-value | Kendall | p-value | Pearson | p-value | Spearman | p-value | Kendall | p-value |
| N. America | -0.3556 | 0.0633 | -0.4143 | 0.0284 | -0.2698 | 0.0453 | -0.3813 | 0.0453 | 0.0066 | 0.9735 | -0.0053 | 0.9844 |
| S. America | -0.4368 | 0.0201 | -0.5216 | 0.0044 | -0.3598 | 0.0068 | -0.3810 | 0.0455 | -0.3492 | 0.0685 | -0.3175 | 0.0177 |
| Eurasia | -0.1693 | 0.3891 | 0.0421 | 0.8314 | 0.0053 | 0.9844 | 0.2640 | 0.1747 | 0.3645 | 0.0565 | 0.2328 | 0.0857 |
| Africa | -0.5061 | 0.0060 | -0.4959 | 0.0073 | -0.3280 | 0.0141 | 0.2405 | 0.2177 | 0.3815 | 0.0452 | 0.2751 | 0.0411 |
| Australasia | -0.4028 | 0.0336 | -0.2151 | 0.2716 | -0.1587 | 0.2463 | -0.2817 | 0.1464 | -0.1598 | 0.4166 | -0.1270 | 0.3564 |

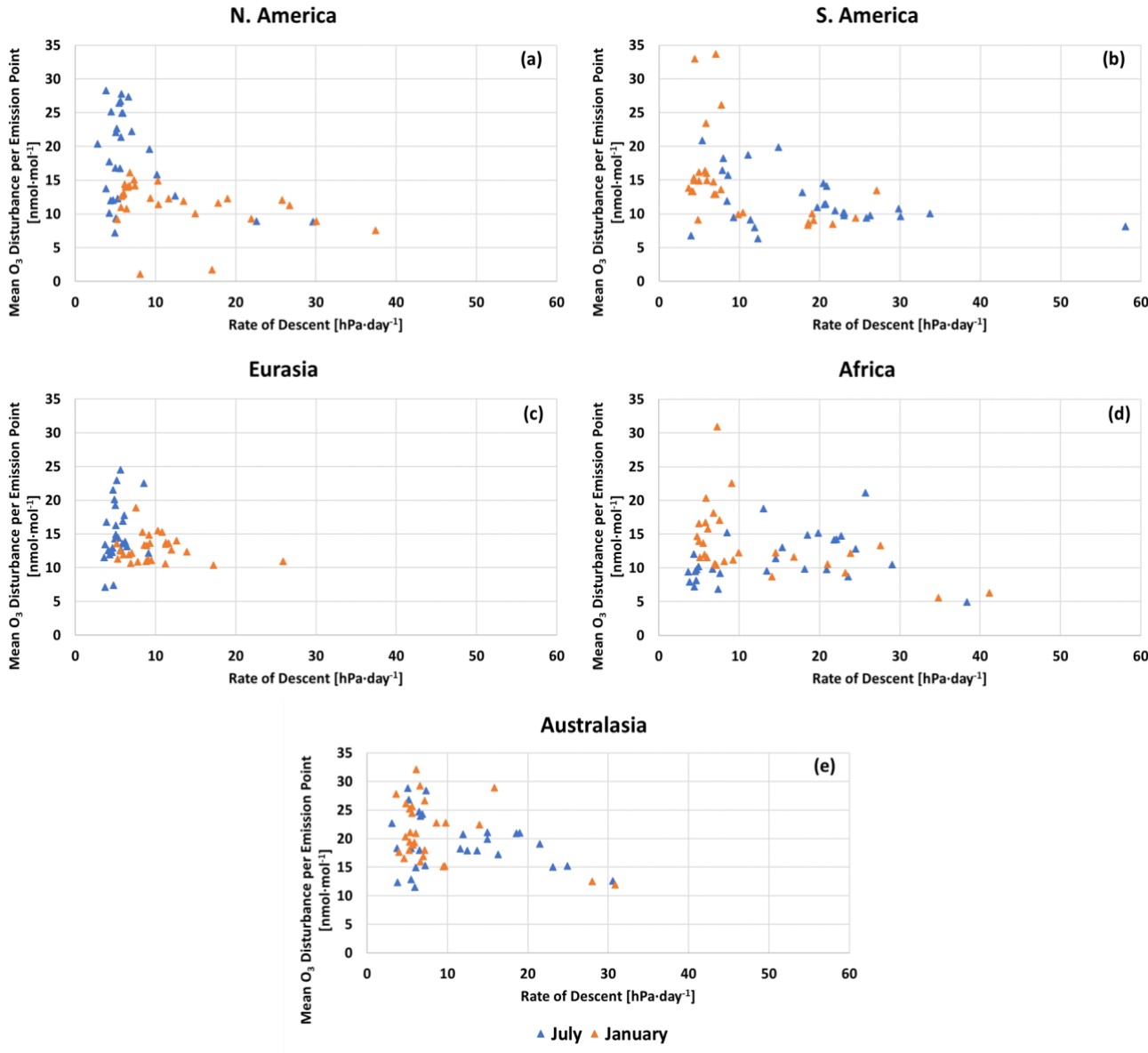

**Figure 6** – Variation of the 90-day rate of descent $\dot{H}$ with the mean $O_3$ disturbance for each of the 28 median $NO_x$-carrying Lagrangian air parcel trajectories for January (orange) and July (blue) for each region.

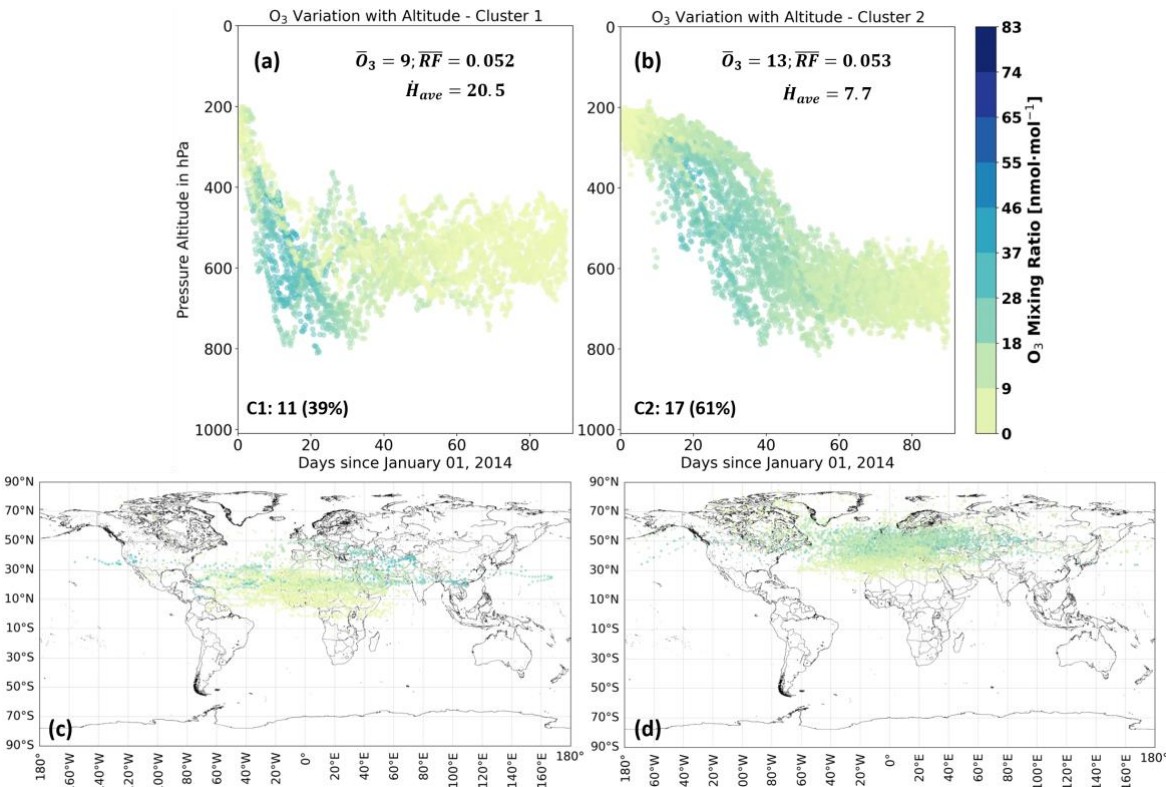

**Figure 7** – (a) Vertical profile of Cluster 1 (C1) of $NO_x$ transport pathways over 90 days following an emission on January 1, 2014 for N. America. The colorbar represents the mean $O_3$ disturbance in nmol·mol$^{-1}$ caused by the median trajectory. Each cluster label is followed by the number of median trajectories that it comprises both in absolute and relative terms (bottom left of each figure). The mean rate of descent for a 90-day period in hPa·day$^{-1}$ for all median trajectories in a cluster is defined as $\dot{H}_{ave}$. The cluster radiative forcing contribution in mW·m$^{-2}$ is labelled as $\overline{RF}$ and the mean $O_3$ disturbance in nmol·mol$^{-1}$ is $\bar{O}_3$. Figure 7 (b) represents the same quantities but for the Cluster 2 (C2), (c) is the horizontal distribution of C1 and (d) is the horizontal distribution of C2.

The link between smaller mean $O_3$ production and an air parcel's faster downward transport is also evident after applying the clustering method, i.e., the two clusters identified by the QuickBundles algorithm for N. America during January – March that are presented in Fig. 7 illustrate the association between descent rates $\dot{H}$ and $O_3$ disturbances. During this season, the vertical pattern of Cluster 2 (C2, Fig. 7 (b)) occurs more often (61%) and is characterized by a slow downward transport until 40-60 days, staying at around 600 hPa. When compared to Cluster 1 (C1, Fig. 7 (a)), however, C2 exhibits a much lower mean descent rate of 7.7 hPa·day$^{-1}$ compared to 20.5 hPa·day$^{-1}$ for C1. The C2 trajectories will therefore remain closer to their original emission altitude for a longer period of time, thereby reducing the $NO_x$ and $O_3$ loss rates and inducing a stronger $\bar{O}_3$ disturbance of 13 nmol·mol$^{-1}$. If the air parcel is immediately transported downwards, as in Fig. 7 (a), the induced $O_3$ disturbance is generally lower, at 9 nmol·mol$^{-1}$, as the emitted $NO_x$ is converted to $HNO_3$ and likely washed out faster. After 40 days, C2 displays an $O_3$ disturbance between $20 - 30$ nmol·mol$^{-1}$ while C1 mostly induces a weaker disturbance of less

than 1 nmol·mol$^{-1}$. Both the mean $O_3$ perturbation and the radiative forcing are slightly larger throughout the 90 days for the scenario of slower descent (C2, Fig. 7 (b)). However, the trajectory with the faster descent rate experiences a larger maximum

$O_3$ of 38 nmol·mol$^{-1}$ compared to the maximum of 34 nmol·mol$^{-1}$ attained by the type of transport of C2. Such a result agrees with findings by Rosanka et al. (2020), who explain that air parcels transporting emitted $NO_x$ will generally attain a larger initial $O_3$ gain if they undergo a fast downward transport, since at lower altitudes the background $NO_x$ and $HO_2$ concentrations are smaller and larger, respectively. This increases $O_3$ production via the reaction initiated between NO and $HO_2$, followed by the photodissociation of $NO_2$ into NO and O. On average however, the smaller lifetimes for $O_3$ at lower altitudes will dominate

and lead to a smaller mean $O_3$ production for C1 compared to C2. Air parcels that remain at higher altitudes, near the tropopause for instance, are subjected to areas in which NO reacts less frequently with $HO_2$, which then leads to lower $O_3$ formation but also smaller $NO_x$ removal rates. One of the determining drivers for the vertical pattern is the vertical wind component, which may be consulted in Fig. C1 (b) and (d) from Appendix C. Regions of intensified downdrafts (e.g., within a subsidence or near gravity waves) will lead to vertical pathways similar to Cluster 1. Figures 7 (c) and (d) show the horizontal distributions for

C1 and C2. It is evident how the trajectories from C1 that have a faster downward transport also travel to lower latitudes compared to C2. Although C1 induces a smaller $O_3$ perturbation, because it travels closer to the Tropics, the RF generated is comparable to C2. The $O_3$ produced by the 50 air parcel trajectories for each emission point for all regions and seasons as well as their consequent RF may be found in the supplementary figures.

Figure 8 presents the individual trajectory residence times of the 1 400 trajectories initialized across the 28 emission points in

each region (e.g., the blue trajectories in Fig. 4) per time period in the following latitudinal bands: Arctic, Northern Midlatitudes and Subtropics, Tropics, Southern Subtropics and Midlatitudes, and the Antarctic. Emissions initialized in the N. Hemisphere (N. America and Eurasia) will mainly reside in the Northern Midlatitudes, spending on average ~39 days between latitudes 35ºN and 65ºN. The tropics are then the second most likely region, with a mean residence time of 25 days. Analogously, for emissions released in the South (S. America, Africa and Australasia), the Tropics display the greatest residence time of 46

days followed by 17 days for the Southern Midlatitudes. The distribution across these latitude bands is similar in both seasons for most cases, with a larger 11% increase in residence time in the Tropics for air parcels initiated in Africa in July relative to January, which is likely explained by the stronger Northern westerly displacing the air parcels mainly to the North (Fig. C1 (a)). For all regions, except N. America during July however, a small southern latitude shift is present.

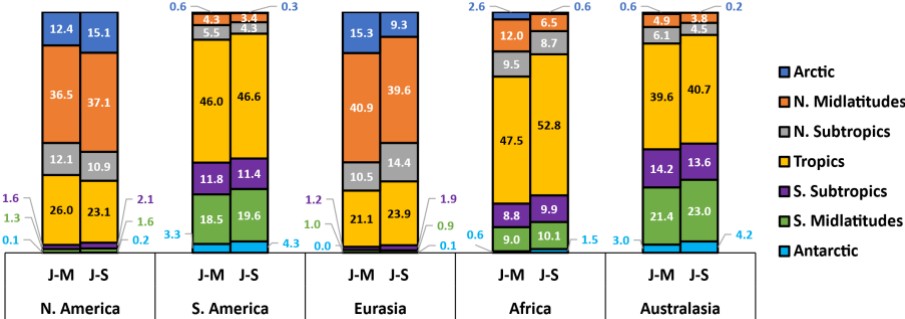

**Figure 8** – Distribution of residence times in days of 1 400 Lagrangian trajectories per emission region and per time period throughout the course of a 90-day path. "J-M" is the period January – March 2014 and "J-S" the period July – September 2014. The latitudinal bands are based on Najibi and Devineni (2018) and are defined as: Arctic = [65ºN,90ºN], N. Midlatitudes = [35ºN,65ºN), N. Subtropics = [23.5ºN,35ºN), Tropics = [23.5ºS,23.5ºN), S. Subtropics = [35ºS,23.5ºS), S. Midlatitudes = [65ºS,35ºS) and Antarctic = [90ºS,65ºS).

The distribution of latitudinal residence times in Fig. 8 also solidifies the reduced tendency for transhemispheric transport and mixing between air parcels released in the North and in the South. Air masses released within the Northern region mainly remain within their hemisphere as more than 70% of their residence time is distributed among the Arctic, Northern Subtropics and Northern Midlatitudes. This does not hold for the Southern regions, where on average slightly more than 50% of trajectories reside in the Tropics for both S. America and Africa, and 45% for Australasia. However, we note that the emission points in S. America, Africa and Australasia are on average closer to the equator than the ones of N. America and Eurasia, which could be driving this. Despite the increased likelihood for transhemispheric transport of trajectories initialized in the South, Fig. 8 generally supports the argument for the limited potential of air parcels to travel to the opposite hemisphere as is dictated by the Brewer-Dobson circulation and verified by earlier multi-model studies like the one by Danilin et al. (1998) as well as more recent ones (Søvde et al., 2014). Other Lagrangian studies (Schoeberl and Morris, 2000; Grewe et al., 2002b) found the same result for current aviation emissions of $NO_y$ (reactive nitrogen) species.

We have seen that the transport behavior and ensuing chemical repercussions vary depending on the meteorological conditions in terms of the existing vertical wind component. The magnitude and direction of the horizontal wind, however, are also drivers. This is evident for emission points 8 and 14 in Fig. 9 from N. America during January 2014.

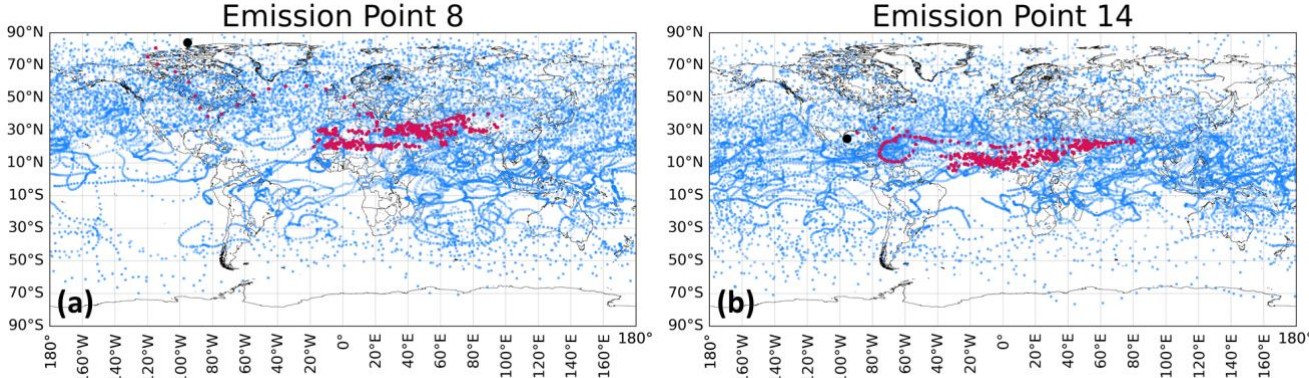

**Figure 9** – Horizontal trajectories for air parcels released in N. America. The black dots denote the emission starting points. Both are located on the Northern Hemisphere in winter (January – March). The red dots represent the median trajectory while the blue dots represent the 50 Lagrangian trajectories from (a) emission point 8 within the meridional tilt and (b) emission point 14 within the westerly and outside of the meridional tilt. The age of the air parcel trajectories is accounted for with variable blue transparency so that ending points appear darker.

Figure 9 illustrates two distinct horizontal transport patterns that are generated as a result of the point of emission relative to the predominant zonal jets. According to the meteorological situation (see Fig. C1 (a) in Appendix C), the Northern region is affected by a strong westerly with a pronounced meridionally tilted jet near the west coast of N. America. Thus, most air parcels starting in N. America, Eurasia and part of Africa will be advected by this jet, as will air parcels emanating from S. America and Australasia by the Southern westerly. If an emission is released well within the westerly and away from the meridional tilt, as is the case of Fig. 9 (b), the ensuing trajectory is mainly contained within the latitudinal bands in which the jet acts during its entire lifetime. However, if the air mass is introduced near the tilt, as is the case in Fig. 9 (a), then it is likely to be transported farther to the South in the direction of the tilt. The vertical and latitudinal evolution of Fig. 9, shown in Fig. E1, also shows distinct patterns. For emission point 8, Fig. E1 (a) first shows a southern shift followed by a pronounced downward motion caused by the air parcel's proximity to an area of stronger downdrafts at 120ºW and 70ºN (Fig. C1 (b)). Emission point 14 is vertically characterized by an almost immediate drop (Fig. E1 (b)) as it is initialized outside of the tilt, directly within a westerly that constrains its latitudinal range and within the same region of stronger downdrafts. From Fig. 9, we also see why the longitude was not used as a representative variable in the clustering function (Eq. (1)). As the trajectories spread and wrap around the globe, they mostly arrive within a similar longitudinal range. For this particular case, the median trajectories mainly end near the prime meridian. The other emission points for N. America for a January $NO_x$ emission may be consulted in Fig. S2. The same plots for the remaining regions may also be viewed in the supplementary materials.

The $O_3$ disturbance and consequent radiative forcing associated with these transport patterns occur along the horizontal trajectories of Fig. 9, as is shown in Fig. 10. A trajectory beginning directly in the Northern westerly (emission point 14) will lead to an increase in $O_3$ production mostly within the latitude band of the jet stream. As such, the radiative forcing contribution is also maximal in the same area. Per contra, an emission in the vicinity of a meridionally tilted jet (emission point 8) will

produce the greatest amount of O$_3$ in a region South of the emission point. Figure 10 (a) identifies the Tropics as the most affected region, i.e., an emission in Canada will likely induce the largest climate impact in Asia, near India and China.

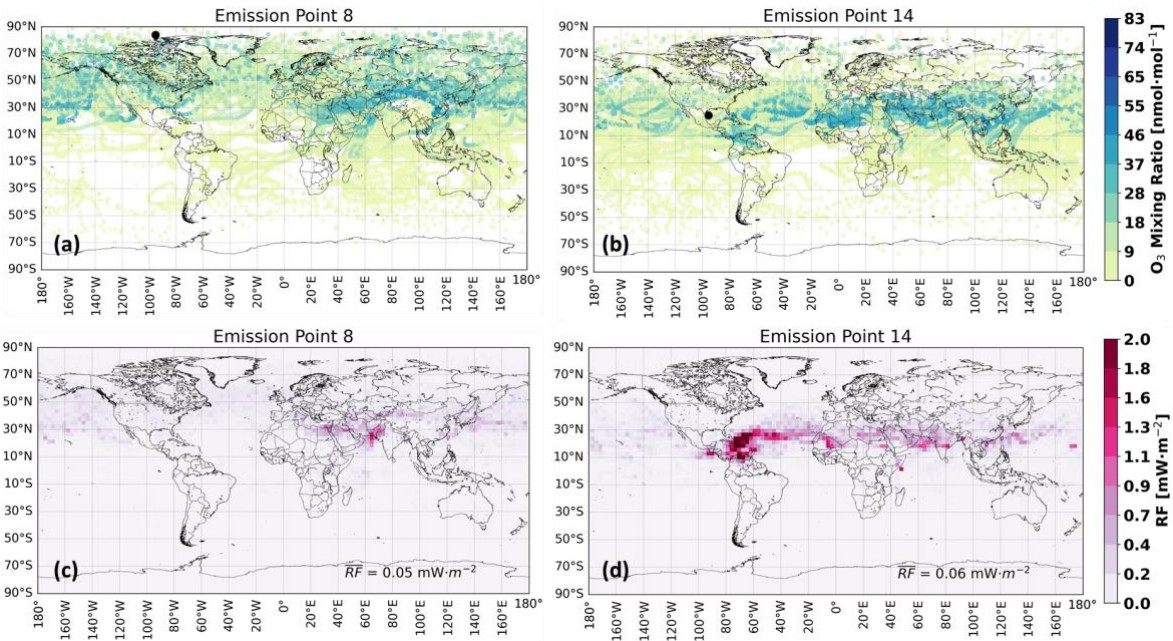

**Figure 10** – Horizontal profiles of mean O$_3$ disturbances in N. America in nmol·mol$^{-1}$ during January 2014 for (a) emission point 8 and (b) emission point 14. The mean radiative forcing $\overline{RF}$ in mW·m$^{-2}$ for (c) emission point 8 and (d) emission point 14. The black dots in (a) and (b) represent the emission starting points.

The seasonal dependence of transport, O$_3$ production, and radiative forcing may be understood by again considering the same emission points as in Fig. 10 and comparing them with the scenario in July 2014, as shown in Fig. 11. When considering the

Northern Hemisphere winter, there is generally a larger O$_3$ production towards the Tropics and the equator (up to 40ºN at most during December – February), as there is amplified solar input. This increased sunlight then leads to more frequent photolysis of NO$_2$ followed by the combination of atomic and molecular oxygen to produce O$_3$. This is consistent with findings by Gauss et al. (2006) and is evidenced in Figs. 10 (a) and (b). Other past studies (Frömming et al., 2021; Köhler et al., 2013) have also verified this tendency for emissions at lower latitudes to generate higher disturbances during the Northern Hemisphere winter

months like January and February. Frömming et al. (2021), for instance, use a similar Lagrangian approach with the EMAC model to conclude that during this Northern winter period, an air parcel emitted within a high-pressure ridge reaches lower altitudes and latitudes along its path and therefore exhibits a larger O$_3$ production, between 30 – 40 nmol·mol$^{-1}$. For a similar time period (Fig. 10 (b)), this study estimates a similar mixing ratio between 30 – 45 nmol·mol$^{-1}$ for the trajectory traveling mainly in the lower latitudes. When comparing radiative forcing values during March 2014 for the North Atlantic flight

corridor, the time-averaged RF per emission point that considers all of the overlapping points between Fig. 1 and their emission

time-region yields the range $[5.3 \times 10^{-3}, 6.9 \times 10^{-2}]$ mW·m⁻². Frömming et al. (2021) estimated the range $[1.2 \times 10^{-2}, 4.7 \times 10^{-2}]$ mW·m⁻² for their stratospheric-adjusted radiative forcing.

During the Northern Hemisphere summer, there is a significant decrease in the intensity of the Northern westerly along with an attenuation of its meridional tilt in N. America according to Fig. C1 (c). The main region of $O_3$ production and radiative forcing for an air parcel starting at emission point 14 (Fig. 11 (b)) remains the same as before: near the Tropics. However, the magnitude of the $O_3$ production is now larger. This result is again consistent with those of Gauss et al. (2006) for N. America during the Northern Hemisphere summer, as the air parcel transported in the North now experiences stronger solar input and in some regions like the Arctic, the exposure to sunlight continues during the whole day. This shift in the exposure to sunlight then translates to a larger $O_3$ production in the Northern mid and high latitudes as is evidenced in Fig. 11, particularly in Fig. 11 (a). As a result of a weaker tilt, the trajectory borne from emission point 8 spends most of its 90-day lifetime in the higher latitudes and is therefore exposed to increased solar radiation, which then generates more $O_3$ (~70 nmol·mol⁻¹ compared to ~40 nmol·mol⁻¹ for emission point 14) and a stronger radiative forcing by a factor of 2 compared to a trajectory released closer to the equator.

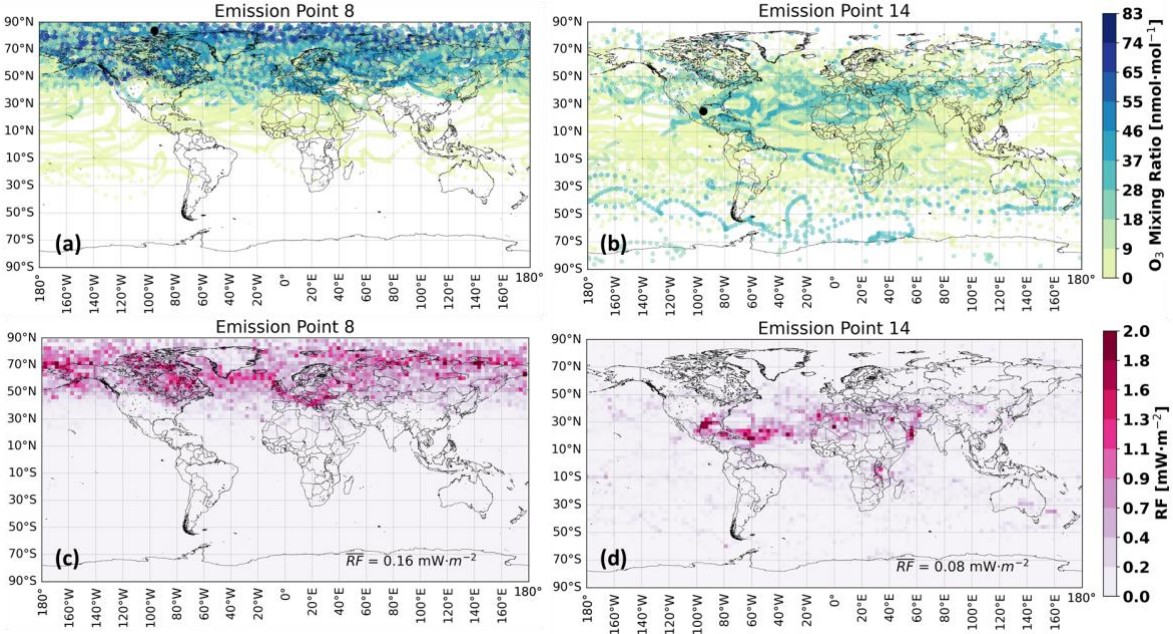

**Figure 11** – Horizontal profiles of mean $O_3$ disturbances in N. America in nmol·mol⁻¹ during July 2014 for (a) emission point 8 and (b) emission point 14. The mean radiative forcing $\overline{RF}$ in mW·m⁻² for (c) emission point 8 and (d) emission point 14. The black dots in (a) and (b) represent the emission starting points.

## 3.2. Profiles of clustered trajectories

Here we present the main transport pathways that result from the $NO_x$ emissions across the 5 regions in Fig. 1 based on the adopted clustering approach with QuickBundles. Their horizontal distributions and vertical profiles for all regions and both time periods are included in Figs. 12 and 13 for emissions in January and July respectively. The $O_3$ disturbance and consequent radiative forcing associated with each of these transport patterns are also shown in the same figures.

### 3.2.1. Profiles for January 2014

The clustering output in Fig. 12 (a) summarizes the vertical profiles of the main transport patterns that were identified for emissions in January. The corresponding horizontal distributions are then depicted in Fig. 12 (b). Each panel in this figure depicts the same information that is shown in Fig. 7: the spatial evolution of a cluster along with its mean $O_3$ and $\overline{RF}$ perturbations.

As shown in Figs. 12 (a) and (b), there are two clusters and therefore two distinct transport pathways for trajectories that are released in N. America at 250 hPa. The vertical profiles (Fig. 12 (a)) depict two different kinds of behavior: C1 shows a faster initial descent when compared to C2. Their rates of descent, $\bar{O}_3$ and $\overline{RF}$ in relation to their vertical behavior have been analyzed earlier in detail in Fig. 7. When considering their corresponding horizontal distributions in Fig. 12 (b), C1 mostly resides closer to the Tropics compared to C2. Consequently, there is more solar input in the region of the $O_3$ disturbance produced by the C1 trajectories. This allows C1 to generate a comparable $\overline{RF}$ value relative to C2, but for a smaller average $O_3$ disturbance of 9 $nmol \cdot mol^{-1}$ compared to 13 $nmol \cdot mol^{-1}$.

The clustering for S. America led to a total of four clusters, with close to 90% of trajectories being categorized in either C1 or C2. The most probable pathway for $NO_x$ released in this region is therefore to follow one of these clusters, especially C1. The vertical profiles for both are similar, where C2 displays a longer average residence time as its rate of descent is lower than in C1. Between these higher-density clusters, the trajectory with the lower rate of descent of 7.4 $hPa \cdot day^{-1}$ (C2) leads to stronger $O_3$ perturbations of 14 $nmol \cdot mol^{-1}$ compared to 11 $nmol \cdot mol^{-1}$ of C1. In terms of their horizontal profiles, C1 travels mainly at higher latitudes than C2. As was discussed earlier, the emissions at lower latitudes during this time period are exposed to more solar radiation and therefore result in a larger radiative forcing. However, the largest radiative forcing and $O_3$ disturbances occur for air parcels in C3 and C4, which have the lowest rates of descent over 90 days.

The transport types identified for Eurasia are similar to those from N. America, where C1 and C2 from Eurasia essentially correspond to C2 and C1 from N. America, respectively. The same conclusion is again possible, where the trajectory with the longest residence time (lowest $\dot{H}_{ave}$) exhibits the highest $\bar{O}_3$. Nonetheless, it appears that the path that descends earlier (e.g., C2 from Eurasia), will lead to a larger maximum of $O_3$ disturbance within the first 20 days upon the $NO_x$ emission, as has been found by Rosanka et al. (2020). Based on the dynamics of all 1 400 $NO_x$-carrying trajectories in Eurasia, the trajectory in C1

is the dominant path with a probability of 75% and is characterized by a slightly lower rate of descent and elevated residence time in the upper Northern latitudes (Figs. 12 (a) and (b)).

For Africa, the most probable path is given by C1, involving almost half of all air masses released (46%). This cluster exhibits the fastest descent and shortest residence time in terms of its $O_3$ disturbance out of all four clusters. The negative correlation between $\dot{H}_{ave}$ and $\bar{O}_3$ is again evident (see Table 1). In terms of the horizontal distributions, C1 resides predominantly near the Tropics, but given its high rate of descent, the $NO_x$ is washed out faster and $\bar{O}_3$ drops under 1 nmol·mol$^{-1}$ earlier than C2. The clusters that induce the largest $O_3$ perturbation are C2 and C4, which also travel mainly through the equatorial region, and additionally benefit from lower $\dot{H}_{ave}$ compared to C1. C3 is the only transport pathway along which the air masses have an initial tendency to be lifted higher than their emission altitude.

Across all regions, Australasia displays the highest $O_3$ disturbances throughout its five clusters and produces the largest radiative forcing. The transport pattern from cluster C3 has the greatest probability of occurrence (64%) and travels mainly in the Southern subtropics, according to Fig. 12 (b), with an approximate rate of descent of 8 hPa·day$^{-1}$. The air parcel with the largest $O_3$ mixing ratio along its path is represented by a single-trajectory, C2, which implies that the likelihood of occurrence is low (~4%). Its $O_3$ maximum occurs when the trajectory is again near its lowest point. The mean RF contributions per region and per cluster for $NO_x$ emissions in January may be found in Table D1.

Overall, the horizontal distributions across the regions are strongly influenced by the atmospheric circulation (zonal jet streams) present in both the Northern and Southern Hemispheres. According to Fig. C1 (a) in the Appendix, the Northern zonal jet is constrained within latitudes 60ºN to 10ºN and the Southern between 30ºS and 70ºS. The former jet splits into two meridionally tilted streams near 150ºW: one is directed towards the northeast and the other towards the southeast. The two clustered horizontal distributions for N. America and Eurasia are each determined by these meridional tilts. The largest northernmost clusters, C2 for N. America (61% of trajectories) and C1 for Eurasia (75% of trajectories), show how the air parcels released in these regions are driven by the north-eastward meridional tilt. The behaviour of the smaller clusters, C1 for N. America and C2 for Eurasia, is influenced by the south-eastward meridional tilts. These clusters therefore lie slightly to the South relative to their larger counterparts. A similar rationale is applicable to the Southern zonal jet as it strongly influences the horizontal distribution of the majority of air parcels released from Australasia, as can be seen by the presence of its highest-density cluster C3 (64% of trajectories) along the latitudinal bounds of this jet.

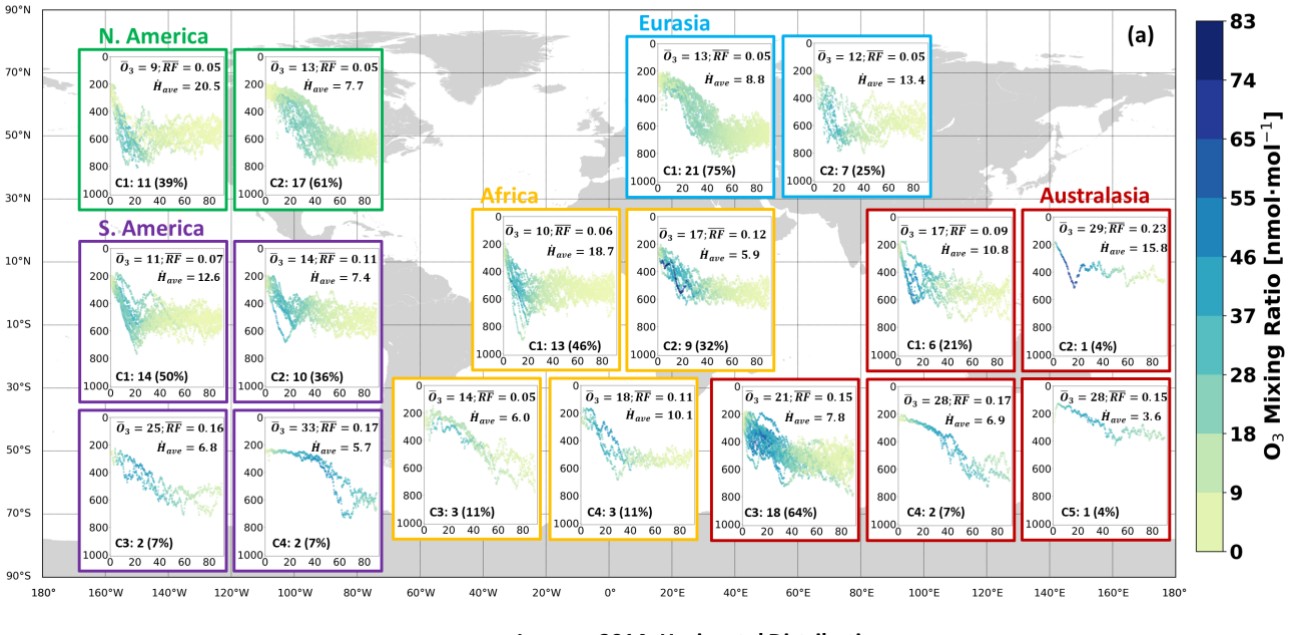

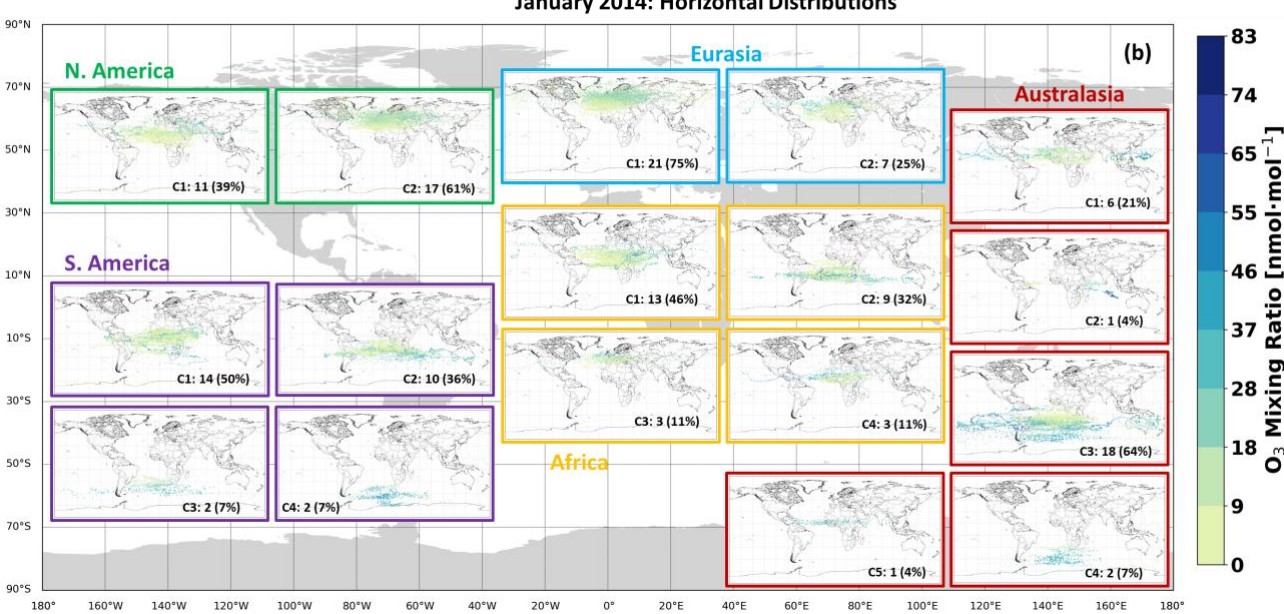

**Figure 12 –** (a) Vertical profile of clusters of $NO_x$ transport pathways during January 2014. The colorbar represents the mean $O_3$ disturbance in nmol·mol$^{-1}$ caused by the emission transported along a median trajectory. Each "C" represents a distinct and independent cluster, i.e., C1 from N. America is not related to C1 from S. America. Each cluster label is followed by the number of median trajectories that it comprises, both in absolute and relative terms. The 90-day mean rate of descent in hPa·day$^{-1}$ for all median trajectories in a cluster is defined as $\dot{H}_{ave}$. The cluster radiative forcing contribution in mW·m$^{-2}$ is labelled as $\overline{RF}$ and $\bar{O}_3$ is the mean disturbance of the median trajectories in nmol·mol$^{-1}$. Figure 12 (b) uses the same notation but for the horizontal distributions. The axes for 12 (a) are defined in Figs. 7 (a) and (b) while the axes for 12 (b) are specified in Figs. 7 (c) and (d). Meteorological conditions considered are specific to the period January 1 – March 31, 2014.

### 3.2.2. Profiles for July 2014

Figure 13 provides the vertical and horizontal distributions of the resulting clusters for all regions and emissions in July. The mean $O_3$ disturbances are now larger in the Northern Hemisphere, i.e., in N. America and Eurasia and on average lower in the Southern Hemisphere. Despite this shift, Australasia continues to globally display some of the largest values in terms of $O_3$ mixing ratios and therefore radiative forcing given the lower background $NO_x$ mixing ratios. In other words, the larger $O_3$ production efficiency, promoted by the cleaner background, increases the amount of $O_3$ produced per unit kg of $NO_x$ emitted, as has been found also by Skowron et al. (2015) and Gilmore et al. (2013).

During July, the most probable path for $NO_x$ emissions in N. America would be C1, with a likelihood of 43% followed by C2 with 36%. C1 also exhibits the highest $O_3$ perturbation and is characterized by a gradual decrease in altitude, according to Fig. 13 (a) at a rate of descent of 6.3 hPa·day$^{-1}$, and remains mainly within the higher northern latitudes (Fig. 13 (b)). For S. America, C2 is the most commonly occurring trajectory type (50%) and also displays the fastest descent at 23.6 hPa·day$^{-1}$ followed by an increase in altitude. Trajectory clusters C1 – C3 for S. America continue to support the tendency highlighted by Frömming et al. (2021) and Rosanka et al. (2020), namely that a faster initial descent will yield larger $O_3$ maxima until being washed out after reaching the minimum in altitude. For Eurasia, all trajectories exhibit similar vertical profiles with relatively low rates of descent when compared to C2 in Eurasia during January – March 2014. The mean $O_3$ perturbations are, as expected, larger than during January, likely due to differences in exposure to solar radiation, and are greatest for C2, which also boasts the largest $\overline{RF}$ in Eurasia among both seasons. For Africa, C2 is the main trajectory (40%), which translates to having most of the emissions quickly descend to lower altitudes and exhibiting the largest $O_3$ maxima along the way, until recovering some of the altitude. Lastly, for Australasia, the most probable path for an air mass is no longer characterized by a gradual vertical decrease like in cluster C3 during January, but now shows a shorter residence time (cluster C2 occurring 46% of the time). The overall decrease in solar radiation towards the lower latitudes is reflected by the lower $\bar{O}_3$ of the air parcel trajectories that are closer to the equator (Fig. 13 (b)). The mean RF contributions per region and per cluster for $NO_x$ emissions in July may be found in Table D2.

Similar to what was stated earlier for the clustering results in Section 3.2.1 for January, it is still clear that the atmospheric circulation strongly influences the geometry of the horizontal distributions shown in Fig. 13 (b) for July. The intensity of the wind field in Figure C1 (c), however, is much smaller for the Northern zonal jet compared to January and the Southern zonal jet is stronger relative to January. Due to this weaker jet stream in the North, not all air parcel trajectories become constrained along the latitudinal band in which they act. Different transport patterns that now reach the lower and mid-Tropics arise for N. America, as can be seen by clusters C2 and C3, together representing almost 50% of trajectories. The Southern zonal jet again strongly influences the transport for S. America (cluster C3) and Australasia (clusters C2 and C3).

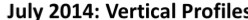

**Figure 13 –** (a) Vertical profile of clusters of NOₓ transport pathways during July 2014. The colorbar represents the mean $O_3$ disturbance in nmol·mol⁻¹ caused by the emission of a median trajectory. Each "C" represents a distinct and independent cluster, i.e., C1 from N. America is not related to C1 from S. America. Each cluster label is followed by the number of median trajectories that it comprises both in absolute and relative terms. The 90-day mean rate of descent in hPa·day⁻¹ for all median trajectories in a cluster is defined as $\dot{H}_{ave}$. The cluster radiative forcing contribution in mW·m⁻² is labelled as $\overline{RF}$ and $\bar{O}_3$ is the mean disturbance of the median trajectories in nmol·mol⁻¹. Figure 13 (b) uses the same notation but for the horizontal distributions. The axes for 13 (a) are defined in Figs. 7 (a) and (b) while the axes for 13 (b) are specified in Figs. 7 (c) and (d). Meteorological conditions considered are specific to the period July 1 – September 30, 2014.

### 3.3. Spatial Variation of Radiative Forcing

This sub-section presents the global radiative forcing associated with emissions in each region (Section 3.3.1) and the source-receptor relationships between them (Section 3.3.2). The radiative forcing shown here represents the difference between the scenario in which an additional $NO_x$ perturbation ($5\times10^5$ kg (NO) per emission point) is introduced and the base scenario in which no aircraft $NO_x$ emissions were added at 250 hPa. As a comparison, this $NO_x$ perturbation in each emission point represents about 40% of the mean total yearly NO emitted by commercial aviation in N. America (as defined by the boundaries

in Fig.1) between the years 2017 – 2020 at a pressure altitude of 250 hPa (Quadros et al., 2022b).

### 3.3.1. Global Radiative Forcing – Most Significant Emitters

The radiative forcing from the emitting regions of Fig. 1 is shown in Fig. 14. An aircraft emitting $NO_x$ near Australasia is most likely to induce a larger (by a factor of 2.7) climate warming during January – March as well as during July – September (by a factor of 1.2) when compared to an emission in Eurasia during the same months. Flying near Eurasia and N. America during

the same time period would, on the other hand, lead to the smallest climate impact globally. The seasonal switch of $\overline{RF}$ between January and July is apparent in all regions. This suggests that, when considering only the short-term $O_3$ increase from $NO_x$, flying over the North Atlantic in January (local winter) will lead to a radiative forcing that is almost half compared to the one induced in July (local summer). In other words, the radiative forcing is larger when flying in their respective summer seasons (or dry season for the Tropics) for all regions.

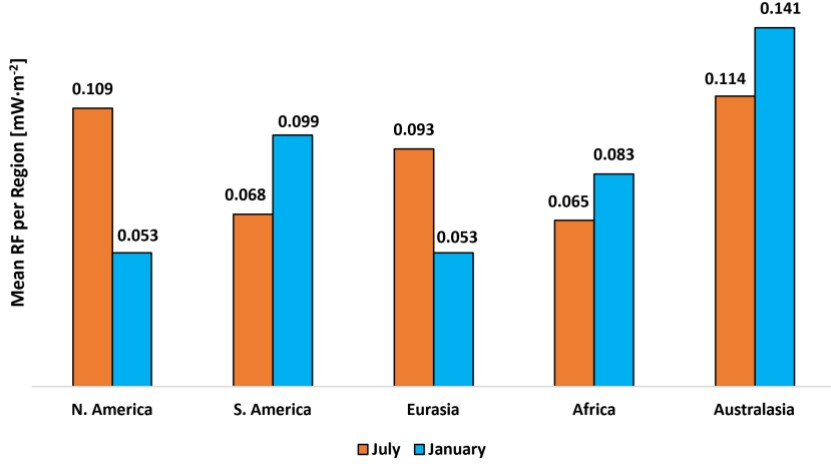

**Figure 14 –** Mean RF in mW·m$^{-2}$ by region of emission during January and July of 2014 for a $5 \times 10^5$ kg (NO) emission.

These results agree with those of Skowron et al. (2015), who also performed a global study (regional breakdown includes Europe, N. America, Southeast Asia, N. Pacific, N. Atlantic, Brazil, South Africa and Australia) with the MOZART-3 CTM to understand the radiative forcing from aircraft $NO_x$ emissions. Cooling effects from $CH_4$, stratospheric water vapor

reductions, long-term O$_3$ considerations (PMO or primary mode ozone), and lastly warming from the short-term increase in O$_3$ were studied. Although Australia did not have the highest net NO$_x$ effect, it did exhibit the largest warming from short-term O$_3$ effects, as is also seen in Fig. 14. The reason for this, compared to other regions, are the larger solar input and reduced background NO$_x$ concentrations in the Australasian region, which then leads to an enhanced O$_3$ production efficiency. Research by Gilmore et al. (2013), using the GEOS-Chem CTM, also helps to justify this result by noting that flights originating from Australia presented the most elevated O$_3$ burdens due to the larger O$_3$ production sensitivities to additional NO$_x$ emissions.

### 3.3.2. Global Radiative Forcing – Regions Impacted (Receptors)

The degree of the impact by emissions released from each of the five source regions (Fig. 1) on the 9 receptor regions defined by Fig. 2 is studied in terms of $\overline{RF}$. The radiative forcing distribution is summarized in Table 2.

**Table 2 –** Source – receptor matrix. Receptor regions defined according to Fig. 2 {NA: North America, NTA: Northern Trans – Atlantic, SA: South America, EU: Europe, AME: Africa & Middle East, AS: Asia, AUS: Australia, NTP: Northern Trans – Pacific, STP: Southern Trans – Pacific}. Results are for a $5 \times 10^5$ kg (NO) emission. The figures in bold represent the maximum RF values attained at a receptor for a given source.

| | **Mean RF [mW·m$^{-2}$]** | **NA** | **NTA** | **SA** | **EU** | **AME** | **AS** | **AUS** | **NTP** | **STP** | **Global** |
|---|---|---|---|---|---|---|---|---|---|---|---|
| **N. America** | **Jan. - Mar.** | 0.112 | **0.113** | 0.014 | 0.109 | 0.033 | 0.086 | 0.005 | 0.105 | 0.007 | 0.053 |
| | **July - Sept.** | 0.205 | 0.220 | 0.039 | **0.295** | 0.056 | 0.178 | 0.020 | 0.186 | 0.025 | 0.109 |
| **S. America** | **Jan. - Mar.** | 0.026 | 0.033 | 0.148 | 0.008 | 0.151 | 0.055 | **0.157** | 0.030 | 0.132 | 0.099 |
| | **July - Sept.** | 0.012 | 0.020 | **0.132** | 0.003 | 0.096 | 0.012 | 0.109 | 0.008 | 0.110 | 0.068 |
| **Eurasia** | **Jan. - Mar.** | 0.101 | 0.123 | 0.008 | **0.132** | 0.026 | 0.087 | 0.004 | 0.113 | 0.007 | 0.053 |
| | **July - Sept.** | 0.192 | 0.190 | 0.020 | **0.218** | 0.042 | 0.128 | 0.014 | 0.217 | 0.019 | 0.093 |
| **Africa** | **Jan. - Mar.** | 0.063 | 0.071 | 0.095 | 0.036 | **0.124** | 0.081 | 0.086 | 0.070 | 0.077 | 0.083 |
| | **July - Sept.** | 0.016 | 0.036 | 0.083 | 0.009 | **0.143** | 0.028 | 0.096 | 0.016 | 0.064 | 0.065 |
| **Australasia** | **Jan. - Mar.** | 0.050 | 0.054 | 0.225 | 0.016 | 0.147 | 0.058 | 0.200 | 0.084 | **0.259** | 0.141 |
| | **July - Sept.** | 0.016 | 0.032 | 0.178 | 0.008 | 0.160 | 0.025 | 0.192 | 0.024 | **0.206** | 0.114 |

Table 2 highlights the notion that the area that will be most affected by an emission will not necessarily coincide with its area of emission. As an example, NO$_x$ emitted in N. America in July will induce the largest climate impact in Europe, with a value of 0.295 mW·m$^{-2}$, the second largest in the Northern Trans-Atlantic (NTA) with 0.220 mWm$^{-2}$, and followed finally by its original emission region, with an $\overline{RF}$ of 0.205 mW·m$^{-2}$. This is likely owed to the fact that NO$_x$-carrying air parcels spend more time during their peak O$_3$ production per unit area within Europe ($5.52 \times 10^{-4}$ hr·km$^{-2}$) than in any of the other receptor

regions (see Table 3). For N. America during an emission on July 1, 2014, Fig. 15 (e) estimates that the maximum $O_3$ production occurs between 14.25 – 20 days after this emission. Compared to the Northern Trans-Pacific region, which bears the second largest $\overline{RF}$ impact, air parcels during this time interval spend 15% more time per $km^2$ in Europe, as is shown by

640 Table 3. This implies that $NO_x$ removal rates, which set the time for the peak location of the $O_3$ production curve, along with the strong westerlies in the Northern Hemisphere, may cause a regional emission to induce a larger $\overline{RF}$ effect farther away from its source. A similar example would be an emission in Australasia in January producing a stronger $\overline{RF}$ signature in S. America (0.225 mW·m$^{-2}$) compared to its own region of emission (0.200 mW·m$^{-2}$). This impact on S. America by Australasia is even stronger than the effect of an emission from S. America itself, with $\overline{RF}$s of 0.132 and 0.148 mW·m$^{-2}$ during July –

645 September and January – March, respectively. This may, in part, be explained by the increased amounts of $O_3$ produced near the Australasian region due to its production efficiency, that are then transported by dominant westerlies to neighboring areas (Fig. C1). A study by Quadros et al. (2020), using the GEOS-Chem CTM, also concluded the existence of pronounced cross-regional impacts for ground-level $PM_{2.5}$ and $O_3$, wherein for instance, an emission in Europe would induce larger impacts in Asia than an emission within Asia itself. It is also worth highlighting again that transhemispheric effects are limited, i.e.,

emissions occurring in the Northern Hemisphere will mainly impact regions in the same hemisphere and the same applies to the Southern Hemisphere.

**Table 3** – Total time spent by all air parcels within a receptor region (Fig. 2) during their maximum $O_3$ peak production period (between 14.25 and 20 days upon emission) as estimated by the $O_3$ production curve for N. America during a $NO_x$ emission in July 2014 (Fig. 15 (e)). This total is then scaled by the respective area to allow for a direct comparison across all receptors.

In these estimates we account for the 1 400 $NO_x$-carrying air parcels released from the emission points in Fig. 1. The percentage values in parentheses represent in relative terms the total time spent by air parcels during peak $O_3$ production for each receptor.

| Receptor | Area (10$^7$ km$^2$) | Total time spent by air parcels during peak $O_3$ production (hr) | Total time spent by air parcels in peak $O_3$ production per km$^2$ (10$^{-4}$ hr·km$^{-2}$) |
|---|---|---|---|
| NA | 7.28 | 34062 (16.9%) | 4.68 |
| NTA | 3.40 | 16344 (8.1%) | 4.80 |
| SA | 9.70 | 7662 (3.8%) | 0.79 |
| EU | 4.32 | 23826 (11.8%) | 5.52 |
| AME | 9.81 | 21528 (10.7%) | 2.2 |
| AS | 8.89 | 47676 (23.6%) | 5.36 |
| AUS | 13.3 | 744 (0.4%) | 0.56 |
| NTP | 9.28 | 47238 (23.4%) | 5.09 |
| STP | 9.70 | 2520 (1.3%) | 0.26 |

## 4 Discussion

The first research question that this study addresses pertains to the identification of the main transport pathways behind gas-phase emitted species ($NO_x$) in the form of clusters across five representative regions (N. America, S. America, Eurasia, Africa and Australasia) and how they are affected by factors like the location of release, seasonality and meteorological conditions. Distinct patterns in emission trajectories were systematically found with the QuickBundles clustering algorithm based on their positional variables (altitude and latitude) and mean radiative forcing. As a second objective, the heterogeneity in the repercussions of different transport patterns on the atmospheric composition of $O_3$ and the consequent radiative forcing was studied. The short-term $\overline{RF}$ of the five emission regions and the main areas of impact are examined from a global, non-clustered perspective. Considering these results, the viability of the QuickBundles algorithm in atmospheric modelling, the main drivers behind the spatio-temporal heterogeneity of short-term $NO_x$ effects and finally, some limitations of the simulation outcomes will be discussed.

QuickBundles adds value to this study by facilitating the identification of transport patterns, while also accounting for their climate impact in a systematic manner. One of the main methodological novelties presented in this study is the integration of this clustering algorithm that enables the consideration of positional and radiative changes per trajectory via a new distance function shown in Eq. (1). Although QuickBundles, in general, has consistently categorized trajectories based on their positional and radiative changes, there are cases in which one or two trajectories appear to be misplaced. One such example is cluster C3 from Eurasia during July in Fig. 13 (a), in which two of the trajectories display a faster rate of descent compared to the rest of the group. This cluster C3 may also appear misleading at first, since it mostly contains trajectories that descend downward earlier, than say cluster C2 with an $\dot{H}_{ave}$ of 5.1 hPa·day$^{-1}$, and yet quantitatively it has a slower average rate of descent ($\dot{H}_{ave}$ of 4.5 hPa·day$^{-1}$). This highlights how the definition given by Eq. 3 may, in cases in which most trajectories reach a value close to their minimum altitude early on and then proceed to gradually and continually descend to slightly lower altitudes until the remaining 90 days, prove to be misleading when comparing vertical transport patterns of median trajectories. As was briefly mentioned in Section 3.1, one possible improvement would be to define an initial rate of descent in which only the first 40 or so days are considered. Another limitation is the small-density clusters comprising a single median trajectory in certain cases like C4 in Africa during July, which would appear to have been better placed in C1 based on the similarities in $\overline{RF}$, $\dot{H}_{ave}$ (Fig. 13 (a)) and their latitudinal range (Fig. 13 (b)). Such shortfalls may be addressed in a future study potentially by introducing weights to each of the three terms of Eq. (1), as was done similarly by Corrado et al. (2020) in their clustering of aircraft trajectories with a weighted distance function. It is worth noting, however, that some single-trajectory clusters like C4 in S. America during July 2014 (Fig. 13 (a)) are justified, given their unique nature relative to the rest of the clusters. The robustness of the clustering also needs to be further investigated in terms of the determination of the optimal cluster number. Kassomenos et al. (2010), in their calculation of the optimal cluster count for the K-means method, stress the volatility of this value based on one's choice of evaluation metric as well as on the method, like trial and error with manual verification. The

optimal value found in this study, which is calculated based on the L-method, is likely to vary if another method is employed,
like the combined use of the sum of squared errors (SSE) with the Silhouette coefficients as was done by Li et al. (2022).
Another important note on the robustness of the clustered results pertains to the strong dependence of the horizontal transport
distributions on the atmospheric circulation, i.e., on the wind field and more specifically on the dominant westerlies in both
the Northern and Southern Hemispheres. This implies that during an event with more extreme weather conditions, the clusters
produced are likely to differ significantly from the ones obtained under more regular meteorological conditions. We do not
believe, however, that our current results were significantly biased by any particular event with more drastic wind conditions.
In 2014, for instance, the El Niño Southern-Oscillation (ENSO) is unlikely to have conditioned the clustering since its
occurrence is considered to have been thwarted that year by a pronounced easterly (Levine and McPhaden, 2016).

The results presented here largely agree with those of other studies that have investigated the short-term effects of the $NO_x$ –
$O_3$ dynamic. A strong spatial dependence of $NO_x$ emissions was found, arising from the differing meteorological conditions at
the location as well as the level of background $NO_x$ concentrations. Frömming et al. (2021), who use a similar simulation setup
also with the EMAC model, found a strong dependence between the ensuing transport pattern and a trajectory's proximity to
areas of elevated subsidence and pressure. The current study also found a similar correlation with emission points 8 and 14
during January (Fig. 9). The former air parcel is initiated within a pronounced meridional tilt of a Northern westerly, while the
latter is directly placed within said westerly but outside of the tilt. From Fig. E1 (a), it is evident that the trajectory consists of
an initial Southern shift resulting from the tilt until it reaches a region of stronger downdrafts near 85ºW and 35ºN, as can be
seen in Fig. C1 (b). The trajectory from emission point 14, however, experiences an almost direct downward motion, as it is
released in the same region of larger subsidence and remains constrained within the latitude band of the Northern westerly.
The significance of zonal jet streams in the transport of pollutants has also been noted by Barrett et al. (2010), with a transport
pattern close to their Fig. 2 arising in Fig. E1 (a). The finding of augmented $O_3$ production and $\overline{RF}$ in areas of reduced
background $NO_x$ concentrations like Australasia (Fig. 14) has also been found in past studies (Stevenson and Derwent, 2009;
Skowron et al., 2015; Gilmore et al., 2013) that have confirmed larger instantaneous RF from $O_3$ and $CH_4$ in areas of lower
background $NO_x$. It was also found that due to the existence of zonal jet streams, the regions impacted by an emission can vary
largely from its origin, as has also been stated by Quadros et al. (2020) in the context of air quality. Table 3 along with Fig. 15
(e) also imply that the time taken to reach the peak $O_3$ production (which is influenced by $NO_x$ removal rates) along with the
total duration spent by $NO_x$-carrying air parcels during this interval may also contribute towards producing an $\overline{RF}$ impact larger
than near the emission source itself.

Seasonality is another driving factor behind the type of transport and consequent $O_3$ disturbances and radiative forcing. This
study finds that, during the Northern Hemisphere winter for instance, there is increased solar radiation in the lower latitudes
near the Tropics, which then translates to larger $\overline{RF}$, as is seen for the Southern regions (S. America, Africa and Australasia)
of Fig. 12. During the Northern Hemisphere summer, this effect is inverted as the largest $\overline{RF}$ occurs for the Northern regions

(N. America and Eurasia), as they benefit from increased solar radiation now in the upper Northern latitudes. This agrees with past studies (Gauss et al., 2006; Frömming et al., 2021; Köhler et al., 2013).

The tagging contribution calculations performed by AIRTRAC rely on the linearization of certain reactions, which means that its output is proportional to the amount of the $NO_x$ emission (Grewe et al., 2014). Additionally, these computations are carried out independently from other emission points and as a result there are no second-order feedback effects considered relative to the background chemistry. This linearization of non-linear chemical processes, such as the $NO_x$-$O_3$ interaction, makes a direct inter-comparison with other models challenging. The background chemistry setup itself, however, has already been evaluated by Jöckel et al. (2006) against observations from 1983 – 2001 of Emmons et al. (2000). Mixing ratios of NO for the simulation year of 2000 were compared to these observations to reveal good skill in the simulated output, having concluded that there is a tendency for the largest $NO_x$ mixing ratios to be underestimated likely due to a lack of $NO_x$ sources and/or an excessively rapid conversion to $HNO_3$. Further validation of the background reaction rates, some of which are used by AIRTRAC, have also been performed by Pozzer et al. (2007). Similarly to Grewe et al. (2014), we compare the resulting $O_3$ disturbance field from AIRTRAC with past research from Stevenson et al. (2004). Figure 15 shows how their results are well constrained within the variation of our $O_3$ mass curves for N. America during January (Fig. 15 (a)) and July (Fig. 15 (b)) and for Eurasia during January (Fig. 15 (c)) and July (Fig. 15 (d)). We also note that large differences may result depending on the method (either contribution or perturbation) chosen to assess aviation's climate effects. A contribution or source apportionment method consists of finding the amount contributed by a certain emission source to the concentration of a chemical species while a perturbation analysis involves estimating the impact of an emission source on the concentration of a chemical species using two simulations: the first contains all emissions (including, for instance, aviation) and the second either switches off this additional emission or reduces its amount. In a linear scenario between emission and resulting concentration, both methods are theoretically equivalent. However, within the highly non-linear context of $NO_x$-$O_3$ chemistry, significant differences are expected (Clappier et al., 2017). Earlier work has shown that if, for instance, the latter method is used to study the contribution of a sector on atmospheric ozone levels, an underestimation by a factor of up to 7 may result for aviation (Grewe et al., 2019) and up to 2.8 for near-surface sectors (Dedoussi et al., 2020). Figure 15 (e) shows the $O_3$ mass temporal evolution averaged across all 28 emission points for each region and time period. Seasonal differences in the $O_3$ production are evident and it is also clear that during January in the Northern Hemisphere, the $O_3$ production peak is lower and occurs later relative to all other scenarios.

In terms of general limitations of this study, the robustness of the results and of the clustering has not been tested for emission altitudes other than 250 hPa, to verify how the heterogeneous $O_3$ production and $\overline{RF}$ vary with altitude. Emissions at other times during the day, other than at 06h00 UTC, would also be interesting in order to assess diurnal effects as well as emitting during other representative days to determine how these results would change for other meteorological situations. A full seasonal analysis with emissions during Spring and Autumn would also be worthwhile. We highlight that our findings are

specific to meteorological conditions from two time periods in 2014: January – March and July – September in 2014. Studying emission points in regions near the Indian Ocean, India and China as well as the North Atlantic may reveal additional transport

patterns. There are also two emission points (21 and 27) from N. America during January in which the 50 air parcels did not equally transport the emitted $NO_x$, likely due to the randomized initialization of the trajectories within the grid cell. Figure 6 (a) depicts these points as outliers. For a more complete assessment needed for mitigation considerations, other non-$CO_2$ drivers should be considered, like the decrease in PMO and $CH_4$, aerosol indirect effects, contrail climate impacts as well as $H_2O$. Lastly, we note that the current resolution of T42L41 was chosen as a compromise between resolution and computational

cost, and a more detailed inspection of the vertical transport for air parcels via a setup with a higher vertical resolution like T42L90MA may provide further insight into the advective and convective impacts on the trajectories across the different regions.

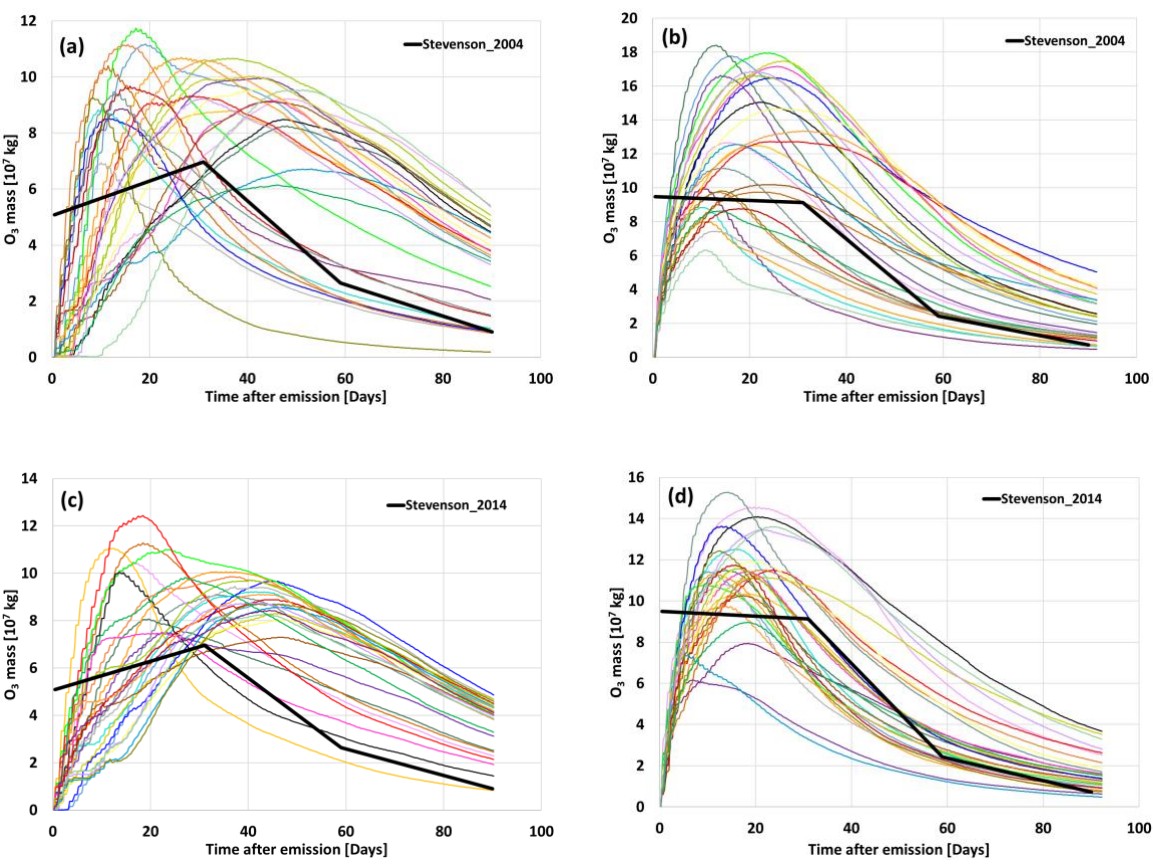

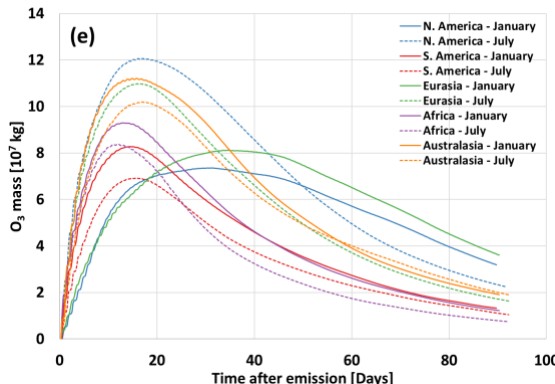

**Figure 15** – The colored lines represent the mean temporal evolution of the $NO_x$-induced short-term $O_3$ mass, in kg, across 50 air parcel trajectories that are initialized in each of the 28 emission points for N. America during January (Fig. (a)) and July (Fig. (b)) as well as for Eurasia during January (Fig. (c)) and July (Fig. (d)). The thicker dark line represents scaled results from Stevenson et al. (2004) to adjust to the same initial $NO_x$ that is released in this study. The scaling factor for a January $NO_x$ emission is $6.97 \times 10^{-3}$ and for a July $NO_x$ emission is $1.22 \times 10^{-2}$. The $O_3$ averaged across all 28 emissions for each region and season is represented in Fig. (e).

## 5 Conclusion

This study assesses the influence of location and seasonality on the transport pathways and climate effects of aircraft $NO_x$ emissions on a global level, using Lagrangian simulations with the EMAC model. The focus has been on the radiative forcing arising from the short-term increase in $O_3$. A clustering algorithm, called QuickBundles, was integrated into the workflow to more efficiently identify the different types of air parcel trajectory groups based on their position (altitude and latitude) as well as on their climate impact ($\overline{O}_3$ and $\overline{RF}$). To the authors' best knowledge, this approach has never been used with GCM results. The outcome of the clustering itself provides a global overview of the possible transport paths that gas-phase emission species may take and their likelihood within the specific meteorological conditions of January – March and July – September 2014.

Results have shown a strong dependence between the location of emission and the ensuing transport type, which is influenced by the wind field. Two dominant zonal jets or westerlies were identified in both, the Northern and Southern Hemispheres, and have proven to dictate air parcel motion along the horizontal direction. An emission occurring in proximity with the meridional tilt of a westerly will tend to travel to the South along the tilt and then stay constrained within the latitudinal band of the jet, while one initialized outside of the tilt will immediately remain constrained to this latitudinal range. Residence times throughout 90-day simulations are also inferred and it is seen that Northern emissions (N. America and Eurasia) spend about 40 days in the Northern Midlatitudes and then around 20 days in the Tropics. For emissions in the Southern regions (S. America, Africa and Australasia), ~50% of the lifetime of the trajectories is spent within the Tropics. Seasonality plays an important role in understanding when certain types of trajectories would have larger $O_3$ perturbations and radiative forcing. During Northern Hemisphere summer (July), air parcels near the higher latitudes will produce more $O_3$ than those to the South,

given the more active photochemistry due to more incoming solar radiation. The opposite is true for winter (January). Globally speaking, $NO_x$ emissions from Australasia are likely to yield the largest climate impact, given their enhanced potential to form $O_3$. A source-receptor analysis has determined that an emission at a given location may induce a larger climate effect onto a neighboring region. Overall, all distinct transport pathways have been summarized per cluster for each region based on their $O_3$ disturbance and $\overline{RF}$ signatures.

The results obtained here have the potential to contribute to climate-optimized aircraft operations by expanding the climate change functions (CCFs). Frömming et al. (2021) have recently produced CCFs for contrail cirrus, aviation-induced $O_3$ by $NO_x$ emissions, total $NO_x$ effects ($O_3$, $CH_4$ and PMO effects) and water vapor for the North Atlantic during the winter and summer seasons. Since the calculation of CCFs is intensive in terms of computational cost, it is not viable to directly integrate them with flight planning tools. To address this shortcoming, the algorithmic CCF (aCCF) was developed (Van Manen and Grewe, 2019). This work therefore provides additional data points that may be applied towards expanding current aCCFs, which focus mainly on the North Atlantic flight corridor, to other regions like Eurasia, Africa, Australasia and S. America.

**Appendix A – Characterization of emission regions**

Air parcel trajectories are initialized at 250 hPa and $NO_x$ emissions ($5 \times 10^5$ kg (NO)) are introduced within a 15-min. time step at each of the coordinate pairs shown in Table A1.

**Table A1 – Emission regions**

| Region | Latitude (ºN) | Longitude (ºE) |
|---|---|---|
| North America | 85º, 75º, 65º, 55º, 45º, 35º, 25º | -115º, -95º, -75º, -55º |
| South America | 10º, 0º, -10º, -20º, -30º, -40º, -50º | -90º, -70º, -50º, -30º |
| Africa | 30º, 20º, 10º, 0º, -10º, -20º, -30º | -20º, 0º, 20º, 40º |
| Eurasia | 60º, 40º | -10º, 0º, 10º, 20º, 30º, 40º, 50º, 60º, 70º, 80º, 90º, 100º, 110º, 120º |
| Australasia | 10º, 0º, -10, -20º, -30º, -40º, -50º | 90º, 110º, 130º, 150º |

**Appendix B – $O_3$ Production and loss terms in the AIRTRAC sub-model**

The net $O_3$ contribution caused by a $NO_x$ disturbance is mainly calculated by adding 3 terms: Net $O_3$ = ProdO3N − LossO3N − LossO3Y. The first term, ProdO3N, refers to the production of $O_3$ that results from Nitrogen species, i.e., $NO + HO_2 \rightarrow OH + NO_2$. The second term (LossO3N) represents $O_3$ loss via Nitrogen species while the third (LossO3Y) is for losses from non-Nitrogen species, i.e., $OH + O_3 \rightarrow HO_2 + O_2$.

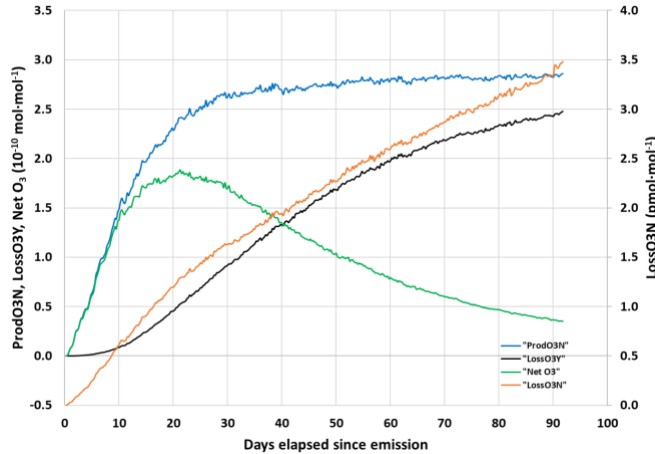

**Figure B1** – Time evolution of O$_3$ production and loss terms for AIRTRAC

## Appendix C – Characterization of meteorological conditions during NO$_x$ emissions

The weather situations during both time periods (July – September and January – March of 2014) are characterized by considering the horizontal wind vector (in ms$^{-1}$, resultant vector from the zonal and meridional wind components, as in Figs. C1 (a) and (c)) along with the spatial evolution of the vertical wind component (also in ms$^{-1}$) and geopotential height (in m) in Figs. C1 (b) and (d), respectively. The geopotential $\phi$ (in m$^2$s$^{-2}$) is obtained directly from EMAC output, while the geopotential height $Z_\phi$ is calculated according to Eq. (C1):

$$Z_\phi = \frac{\phi}{9.80665} \tag{C1}$$

In all cases, an averaged snapshot of the first 18 hours, at an approximate altitude near the emission altitude (250 hPa), is shown to better portray the initial conditions.

Relative to the global weather pattern for January – March 2014, which is detailed in Figs. C1 (a) and (b), dominant zonal jets with intensities above 60 ms$^{-1}$ are found in the latitude bands from 60ºN to 10ºN and 30ºS to 70ºS in the northern and southern hemispheres, respectively. Pronounced meridional tilts exist in both cases, particularly in the west between 150ºW and 100ºW. The northern jet stream splits into two westerlies at around 150ºW. Near the tropics, the horizontal wind is at its weakest, with only minor trade winds being detectable.

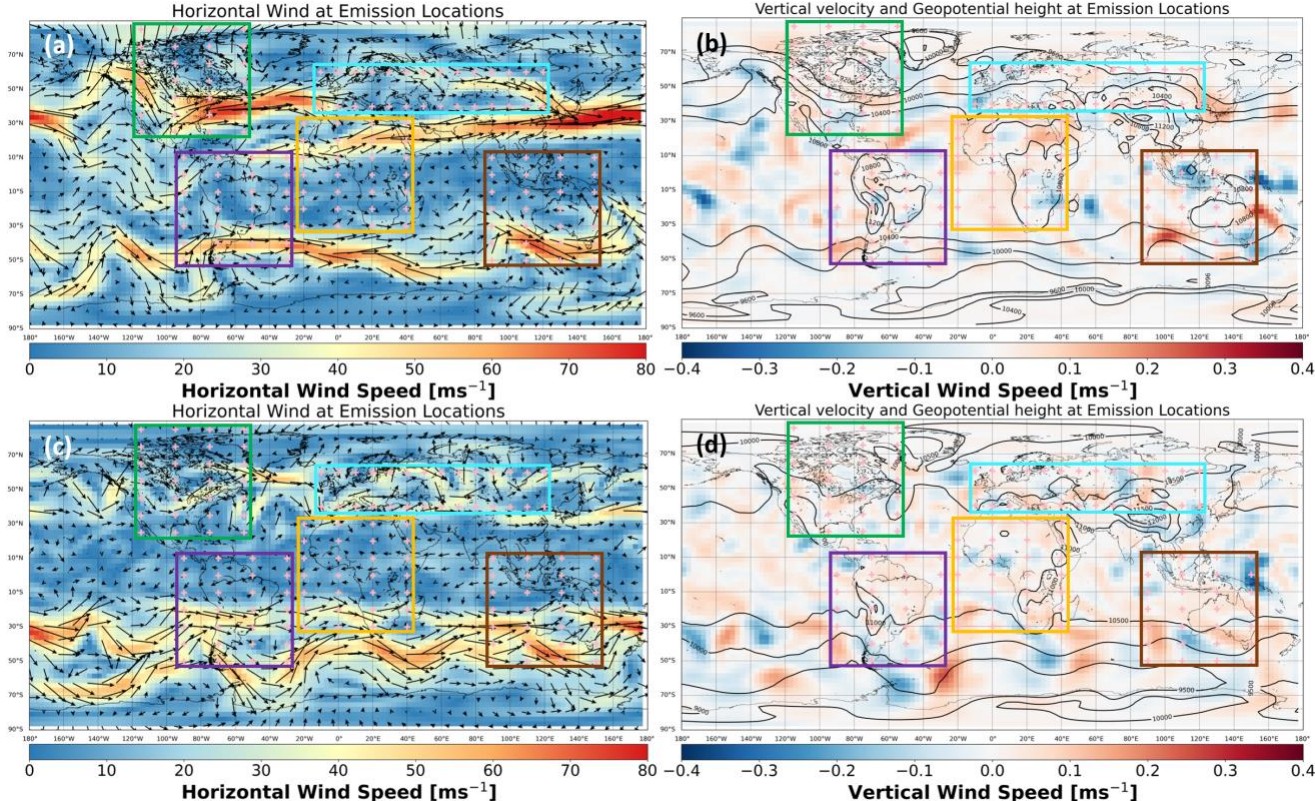

**Figure C1 –** Averaged horizontal and vertical wind fields at ~250 hPa during January – March 2014 in Fig. C1 (a) and C1 (b). Same information is portrayed in Fig. C1 (c) and (d) for July – September. Geopotential height contours are expressed in m. The same emission points as in Fig. 1 are shown with "+" in each region.

Contours of higher geopotential height from Fig. C1 (b) typically envelope a region with stronger downdrafts, as is shown by the negative vertical velocity component near the west coast of S. America for instance. The Atlantic region contains an area of alternating vertical velocities that spans the east coast of N. America to the east coast of S. America, which could indicate a region governed by stronger gravity waves (e.g., vertically propagating waves). The area near Indonesia is characterized by stronger downdrafts of magnitude above 0.20 ms$^{-1}$. Australia's east coast also showcases a pair of strong up- and downdrafts of similar magnitudes: 0.40 ms$^{-1}$. Similar patterns were identified by Frömming et al. (2021), who focused on the North Atlantic region. In both studies, strong zonal jets were found during the winter period, especially in their weather patterns W1 and W2 in their Fig. 2.

The weather pattern for the period of July – September 2014 is shown in Fig. C1 (c) and (d). The horizontal wind (Fig. C1 (c)) is similar to the pattern of Fig. C1 (a) with two zonal jet streams at the Midlatitudes. During July, the southern westerly is more pronounced than in the North, while the opposite is true for January. Slightly stronger trade winds are visible near India. The Southern Midlatitudes that comprise the strongest westerly also exhibit a prominent alternating pattern of up- and downdrafts,

 which points to the possibility of accentuated wind shear and turbulence. At approximately 10ºN, several occurrences of downdrafts are present. Emission point 1 from S. America, for instance, is placed directly in a region of descending air masses and so are several emission points from Australasia like 1, 3, 13, 22 and 23. The summer weather patterns S1 – S3 by Frömming et al. (2021) similarly exhibit a weaker zonal jet in the North Atlantic compared to winter.

**Appendix D – Mean Radiative Forcing per Cluster**

The estimated mean radiative forcing ($\overline{RF}$) impact for each cluster is tabulated for January – March and July – September in Tables D1 and D2 respectively, along with a 95% confidence interval.

**Table D1 –** Mean radiative forcing impact in mWm$^{-2}$ normalized by trajectories per cluster during January – March 2014. The rows denote the clusters in each region.

|    | N. America | S. America | Eurasia | Africa | Australasia |
|----|-----------|-----------|---------|--------|-------------|
| C1 | 0.0521 [0.0498, 0.0545] | 0.0733 [0.0711, 0.0756] | 0.0525 [0.0510, 0.0545] | 0.0590 [0.0570, 0.0611] | 0.0905 [0.0867, 0.0943] |
| C2 | 0.0529 [0.0513, 0.0545] | 0.1061 [0.1025, 0.1098] | 0.0529 [0.0512, 0.0545] | 0.1185 [0.1139, 0.1230] | 0.2253 [0.2093, 0.2413] |
| C3 | ---- | 0.1641 [0.1576, 0.1705] | ---- | 0.0545 [0.0527, 0.0563] | 0.1492 [0.1443, 0.1540] |
| C4 | ---- | 0.1745 [0.1671, 0.1820] | ---- | 0.1130 [0.1090, 0.1170] | 0.1720 [0.1650, 0.1790] |
| C5 | ---- | ---- | ---- | ---- | 0.1454 [0.1391, 0.1517] |

**Table D2 –** Same as Table D1 but for July – September 2014.

|    | N. America | S. America | Eurasia | Africa | Australasia |
|----|-----------|-----------|---------|--------|-------------|
| C1 | 0.1348 [0.1301, 0.1395] | 0.0769 [0.0735, 0.0803] | 0.0941 [0.0910, 0.0972] | 0.0528 [0.0509, 0.0546] | 0.1456 [0.1412, 0.1501] |
| C2 | 0.0672 [0.0654, 0.0690] | 0.0615 [0.0594, 0.0637] | 0.1180 [0.1141, 0.1219] | 0.0685 [0.0658, 0.0712] | 0.1070 [0.1032, 0.1107] |
| C3 | 0.1425 [0.1379, 0.1471] | 0.0773 [0.0747, 0.0800] | 0.0772 [0.0746, 0.0798] | 0.0816 [0.0789, 0.0843] | 0.1049 [0.1012, 0.1086] |
| C4 | 0.1138 [0.1098, 0.1178] | 0.0299 [0.0285, 0.0313] | 0.0984 [0.0953, 0.1015] | 0.0534 [0.0502, 0.0566] | 0.1162 [0.1119, 0.1205] |

**Appendix E – Vertical transport of NO$_x$ for emission points 8 and 14 of Figure 9**

Figure E1 is a depiction of the transport of emitted NO$_x$ in terms of its latitude and altitude, released at two emission points (8 and 14) that are discussed in Figs. 9 – 11.

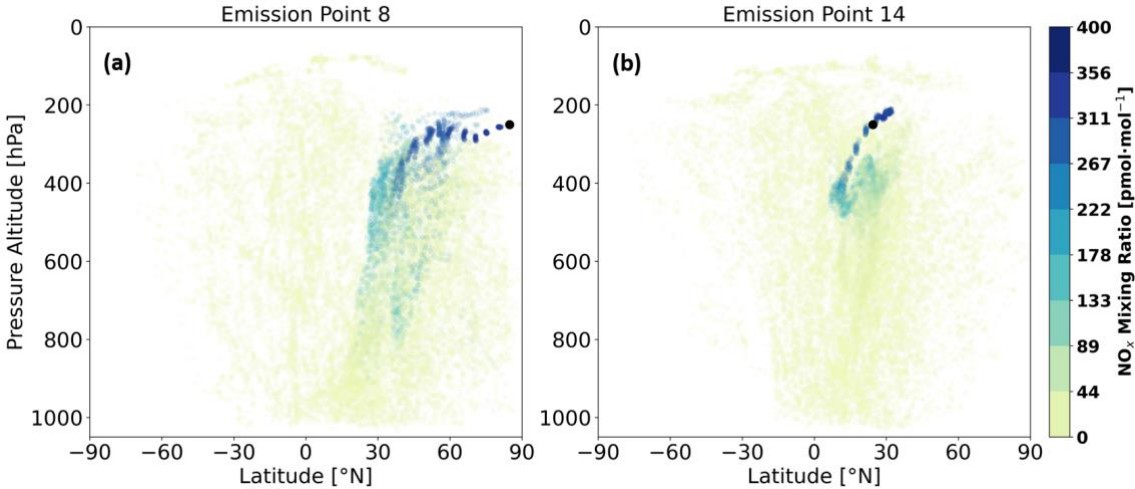

**Figure E1** – Transport of emitted NO$_x$ released in N. America during January 2014 at (a) Emission Point 8 within a meridional tilt and at (b) Emission Point 14 outside the tilt but within the Northern westerly. The black dot denotes the emission point. The NO$_x$ mixing ratio is expressed in pmol·mol$^{-1}$.

**Acknowledgments**

We would like to express our gratitude to Dr. Sabine Brinkop from DLR (German Aerospace Center), who was kind enough to share her expertise regarding the EMAC sub-models, particularly with ATTILA and to Dr. Alina Fiehn, also from DLR, for providing an internal review.

This work was carried out on the Dutch national e-infrastructure with the support of SURF Cooperative.

**Funding**

This research is part of the ACACIA (Advancing the Science for Aviation and Climate; www.acacia-project.eu) Project, which is funded by the European Union under the Grant Agreement No. 875036.

**Code and Data Availability**

The Modular Earth Submodel System (MESSy) is continuously further developed and applied by a consortium of institutions. The usage of MESSy and access to the source code is licenced to all affiliates of institutions which are members of the MESSy Consortium. Institutions can become a member of the MESSy Consortium by signing the MESSy Memorandum of Understanding. More information can be found on the MESSy Consortium Website (http://www.messy-interface.org, accessed on November 1, 2021). The code presented here has been based on MESSy version 2.55 and will be available in the next official release (version 2.56).

The QuickBundles clustering code is openly available at: https://dipy.org.

EMAC simulation output data that was produced and analyzed in this paper is openly available at https://doi.org/10.4121/16886977.

## Supplement

Supplementary figures are included in the following data repository: https://doi.org/10.4121/20338212.

## Author Contributions

JM, VG, and ID contributed to the conceptualization of the study. JM, VG and ID performed the analysis of simulation results. VG and PJ contributed with overall support for the installation of EMAC and its sub-models on the Dutch supercomputer and for creating the specific model setup. CF provided complementary analysis tools for the simulation and support for some sub-
models. JM produced the manuscript with input from all authors.

## Competing Interests

Co-author Patrick Jöckel is a member of the editorial board for the atmospheric modelling subject area.

## Disclaimer

Boundaries in Figs. 1 and 2 are arbitrarily defined and therefore have no political significance.

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
