# Peer review of "Transport Patterns of Global Aviation NOx and their Short-term O3 Radiative Forcing – A Machine Learning Approach"

_Atmospheric Chemistry and Physics, 2022_

## Author Response (AR1)

Dear Editor,

Re: Submission of manuscript titled "Transport Patterns of Global Aviation $NO_x$ and their Short-term $O_3$ Radiative Forcing – A Machine Learning Approach" to *ACP*.

We thank the referees for the insightful comments on our manuscript. Below, you will find our response to the referees, answering each comment and detailing how their suggestions were incorporated in the revision (referee comments in *italics*, our response **in bold** font, page/line numbers referring to the updated manuscript). We have also provided a document highlighting the changes to our manuscript.

Overall, we have provided more detail regarding our adopted methodology and the reasoning behind it, expanded the interpretation of some of our key results regarding the source-receptor analysis, and refined the visualization of some of our results. We have also expanded the literature presented on aviation $NO_x$-$O_3$ chemistry. Finally, further discussion on some of the limitations of our study (e.g. the consideration of a single year of meteorology) has also been included. We believe that these changes have improved the quality of our manuscript.

Thank you again for considering our submission to *ACP*, and we look forward to your response.

Sincerely,

Irene Dedoussi

**Anonymous Referee #1**

*General Comments*

*The paper titled "Transport Patterns of Global Aviation NOx and their Short-term O3 Radiative Forcing – A Machine Learning Approach" by Maruhashi et al. explores the pathways traced out by aviation NOx emissions and the effect this has on radiative forcing, including spatial variations. Overall, the paper is well written with a logical structure and makes a solid contribution to the field. The introduction of a new method borrowed from neuroscience can benefit the field of climate research and is of interest to many ACP readers, particularly given how well it works when applied to NOx pathways here.*

*Readers will appreciate that the introduction gives an overview of the field currently, including where similar methods have been used in the past, making the novelty of this work clear. The methods are generally well described. Addressing some of my comments below will make the methods even clearer to the reader.*

*The analysis of the results is very in depth and there are multiple new contributions made in the paper. The authors show the pathways of NOx in both horizontal and vertical directions as well as the evolution of O3 production, which are useful contributions to the community. In the last results section, the authors present the radiative forcing, including both local and remote effects of NOx emissions with a particularly interesting finding that the largest radiative forcing associated with regional NOx emissions are not always co-located near the source (e.g. Eur is the strongest receptor of RF given NAm source emissions in Jul-Sept. in Table 2). These results are of interest to the community, although they would benefit with some expansion and more with reference to the atmospheric circulation of the time to explain whether the results would hold year to year. Overall, the findings are backed up with relevant references throughout. Some minor revisions will make this paper ready for publication.*

We would like to thank the referee for their time in providing constructive feedback that has improved our manuscript. We are also pleased to read that our research is deemed a relevant and novel contribution to the ACP journal audience.

*Specific Comments*

*Section 2.1: it was not clear to me if all 28 trajectories are released in independent simulations or within the same simulation. If it is the same simulation*

**The trajectories associated with the 28 emission points for each region are indeed released in the same simulation. We have added a sentence to clarify this (lines 166 – 168):**

**"With this Lagrangian approach, a single simulation is sufficient to calculate the contribution of a $NO_x$ emission to atmospheric $O_3$ for all 28 emission points in parallel for a given region based on the 50 emission-carrying trajectories that are released at each of these emission points (Grewe et al., 2014)."**

*L232: "instead of the longitude which exhibits less variation between the trajectories with each region." This sentence confused me initially as it is not yet clear that the clustering is done within each region. Some rewording could make this clearer.*

**We have rephrased the sentence (now in lines 262 – 265) to:**

**"However, this metric has been adapted in our study to account for the impact of $NO_x$ emissions on the Earth's energy balance during the clustering process by replacing the longitude of the air parcels with the radiative forcing, where the former variable exhibits less variation between the trajectories in each region than the latter."**

*L259: "It consists of finding the intersection between two linear regressions that model each half of the curve generated from plotting the evolution of a trajectory-based metric as a ..." This does not explain the method well. The method becomes clearer when we are introduced to Fig. 5 so some reshuffling may be needed.*

**We agree that this method could benefit from further explanation. We have added additional details in Section 2.4.2 (lines 299 – 311). We have also reshuffled the figures such that the related figure (currently Figure 4) appears first. This way, the description of the method is closer to the relevant figure, in order to help the reader better understand the L-method.**

*L285: "The rate of descent". Could you explain further the relevance of this quantity, why do we care about it? How do we expect it to relate to O2 production? Is this a commonly used quantity in other studies of aviation NOx? Is it always defined in this way? Is there a benchmark value that we can expect this quantity to be from previous studies?*

**We have defined the "rate of descent" of air parcel trajectories as the ratio between the vertical distance from their point of release (250 hPa) to the minimum altitude they reach throughout the 90-day simulation divided by the time taken for this to occur. It can be equated to a vertical average speed until the air parcel reaches its minimum altitude. This quantity is of interest to us because we want to verify if, for most regions, there is a relation between how quickly an air parcel was transported downwards and its mean $O_3$ production. By observing Figs. 6, 7 (a) and 7 (b) which show the clustered North American vertical profiles for January 2014, we suspected that a faster downward transport (larger $\dot{H}$) was associated with a lower mean $O_3$ production. This could be explained by faster removal rates for $O_3$ (therefore shorter lifetimes) at lower altitudes compared to**

**a more efficient accumulation at higher altitudes. Similar findings are shown in Fig. 4 from Grewe et al. (2002)[1]. To statistically confirm this hypothesis we then calculated three correlation statistics: Pearson, Spearman and Kendall, which we present. To the best of our knowledge, it is a metric that is first defined here, so we have expanded a sentence in Section 2.5 (lines 334 – 336) emphasizing the purpose of H:**

**"The quantity Ḣ is […] intended as a metric to understand if the initial downward transport has an impact on the atmospheric concentrations of $O_3$."**

*L315: "Table 1 Statistical correlations using the Pearson, Kendall, ..." Should be "statistical correlations between H and mean O3 contribution using the .... "*

**We have implemented this change.**

*Fig. 7 and other similar plots: I would suggest a darker color scheme, even just shifting this slightly to lower O3 mixing ratio as the pale yellow does not appear too clearly. I would also recommend to reduce opacity of each point (e.g. in python, alpha = 0.5) if this has not already been applied.*

**We have chosen to keep the same color scheme, but have removed the lightest shade of the original colorbar (subset of original colors) to improve readability by utilizing more of the darker colors. We have also changed the alpha setting to 0.5 as suggested. Figures 10 – 13 have also been updated accordingly.**

*Fig. 9 and other similar plots: For this and other similar figs. there is a lot of information in the green dots, perhaps making different trajectories different shades of green and/or slightly transparent would help. Alternatively, the time series of the trajectories could be shown graphically, e.g. so that starting points on the trajectory are lighter and ending points are darker, which should verify L395 "As the trajectories spread and wrap around the globe, they mostly arrive within a similar longitudinal range". Also, for Fig. 9, please check whether the red points are visible against the green for people with red-green color blindness.*

**We have updated Figures 5, 9, S1-2, S7-8, S13-14, S19-20, S25-26, S31-32, S37-38, S43-44, S49-50, and S55-56 from the supplementary materials so that the age of the air parcel trajectories are discernible based on the variation of the blue transparency setting. Their colors have also been updated so that they are colorblind-safe.**

*Section 3.2: There could be more explanation of the results of Fig. 12 and 13 based on the atmospheric circulation in Fig. C1. Do we expect these to hold year to year? Are there particular conditions e.g. ENSO that occurred in 2014 that may bias these results?*

**We have included additional discussion in Sections 3.2.1 and 3.2.2, relating to Figures 12 and 13 respectively, on the effects that the atmospheric circulation may have on the resulting transport paths.**

**Fig. C1 (a) is now mentioned in Section 3.2.1 to explain how the dominant Northern zonal jet stream during January 2014 influences the nature of the transport pattern in the horizontal (latitude-longitude) evolution of the air parcels released in North America and Eurasia. The splitting of this jet stream into two meridional jets is responsible for creating two distinct clusters for both of these**
* * *
[1]Grewe, V., Dameris, M., Fichter, C., and Lee, D.S.: Impact of aircraft $NO_x$ emissions. Part 2: Effects of lowering the flight altitude, Meteorol. Z. 3, 197-205, 2002.

regions: clusters C1 in North America and C2 in Eurasia are influenced by the meridional jet pointing to the North while clusters C2 in North America and C1 in Eurasia are affected by the Southbound meridional jet. The Southern zonal jet stream that is found between 30ºS to 70ºS is also responsible for dictating the horizontal behaviour of most air parcels released from Australasia seeing as the highest-density cluster C3 (representing 64% of trajectories) is present along the bounds of the zonal jet. An additional paragraph has been added in Section 3.2.1 to discuss these ideas (lines 542 – 552).

Fig. C1 (c) is also now mentioned in Section 3.2.2 to similarly discuss the impact of the atmospheric circulation on the horizontal distributions of the various clusters. The Northern zonal jet is now no longer as strong in terms of wind speed as it was during January 2014, however, it is still able to constrain most of the air parcel trajectories within the latitudinal range at which it exists. The Southern zonal jet is now stronger than it was in January and is therefore still the dominant factor in determining the geometry of the horizontal distributions of the dominant paths for the Southern regions, especially S. America and Australasia. An additional paragraph has been added at the end of Section 3.2.2 (lines 585 – 591).

In addition, we have noted that the transport patterns and their clustering are influenced by the zonal jets whose intensities vary throughout the year. We have also seen, via the comparison of Figs. 12 and 13, how varying the magnitude of the wind field can create different types of transport patterns, thereby also having an impact on the clustering. So when comparing throughout the year, the transport patterns will indeed vary. When considering the same time period on a year-to-year basis (e.g. Jan. – Mar. 2013 vs. Jan. – Mar. 2014) however, we do not expect significantly different results given the approximately constant mean of the wind speed for jet streams present between 100 and 400 hPa. Based on Fig. 3 of Archer and Caldeira (2008)[2], we would expect more variability in the Southern transport patterns given their increasing trend in wind speed. Generally speaking, the westerly winds in both Hemispheres are responsible for shaping the transport patterns across all regions. Given their prevalence throughout the year[3], we believe that the clustering results should not vary drastically.

Finally, for specific phenomena that may significantly affect atmospheric conditions like the El Niño Southern Oscillation (ENSO), it is possible that the transport patterns would be quite disparate. For the 2014 ENSO event, however, we do not believe that there would be a significant bias for our results since a large easterly wind was responsible for stopping the El Niño event in 2014 (Levine and McPhaden, 2016)[4]. We have summarised this discussion in Section 4 (lines 691 – 694):

"We do not believe, however, that our current results were significantly biased by any particular event with more drastic wind conditions. In 2014, for instance, the El Niño Southern-Oscillation (ENSO) is unlikely to have conditioned the clustering since its occurrence is considered to have been thwarted that year by a pronounced easterly (Levine and McPhaden, 2016)."

*L533: "This suggests that flying over the North Atlantic in January (local winter) will lead to a radiative forcing that is almost half compared to the one induced in July (local summer). In other words, the radiative forcing is larger when flying in their respective summer (or dry season for the Tropics) seasons for all*

[2]Archer, C. L., and Caldeira, K. (2008), Historical trends in the jet streams, Geophys. Res. Lett., 35, L08803, doi:10.1029/2008GL033614.

[3]http://www.ces.fau.edu/nasa/content/resources/global-wind-patterns.php, accessed on Sept. 12, 2022.

[4] Levine, A. F. Z., and McPhaden, M. J.: How the July 2014 easterly wind burst gave the 2015–2016 El Niño a head start, Geophys. Res. Lett., 43, 6503– 6510, 2016, doi:10.1002/2016GL069204.

*regions." A rather bold statement, worth reminding the reader that this due to NOx only, not including GHG contributions? Also move "seasons" to before "(or dry season for the Tropics)".*

**We agree that it is worth reminding the reader here that this statement only takes into account the short-term $O_3$ produced by $NO_x$ emissions. We have clarified this in the relevant sentences (lines 612 – 615).**

*Section 3.3.2: The findings here are the most interesting of the paper. Can you explain why Europe may be most impacted by N. America source emissions in summer. Given the detail in the earlier sections of the paper I would like to see more explanation of this, as the tracer profiles are already available and we can see where the emissions end up e.g. Fig. 9., 12 and 13.*

**These relationships are influenced by various parameters: the local time of emission, the meteorology, the area of each receptor region, as well as the background atmospheric composition along the trajectories (including ozone and climate sensitivity). In order to assess any impact that the total amount of time spent by $NO_x$-carrying air parcels in each receptor region has, we calculate and discuss the total amount of time spent by $NO_x$-carrying air parcels specifically during their maximum $O_3$ production (between 14.25 and 20 days upon emission) averaged by the area of each receptor region. This is included in the newly added Table 3. Overall, we find that $NO_x$-carrying air parcels spend more time during their peak $O_3$ production per unit area within Europe ($5.52 \times 10^{-4}$ hr·km$^{-2}$) than in any of the other receptor regions, which could be the reason of the higher RF. The corresponding discussion is included in lines 636 – 643. An additional sentence summarizing this explanation was also added in the Discussion section (lines 710 – 713).**

*Please also swap the order of the table months so that Jan-Mar comes above JulySept. to coincide with the order of figures as well.*

**The order of the months in Table 2 has been updated. The values within this table have also been rounded to 3 decimal places to be consistent with those of Fig. 14 in terms of the number of significant figures.**

**Anonymous Referee #2**

*Overview*

*The study in this manuscript investigates the impact of aircraft nitrogen oxides (NOx) emissions on short-term ozone (O3) changes and related radiative forcings. The source-receptor analysis is investigated using clustering of trajectories using the QuickBundles utility which has originally been developed for neuroscience. The paper is organised in a logical fashion, and generally well written. The description of the methodology is fairly clear and seems appropriate; it brings in a few aspects a novel approach to the community for whom this manuscript is likely to be of interest. The study produces a wide range of results which have significant complexity. The authors have attempted in their analysis to present the most important results with lucidity. I have various general and specific comments which I feel would help further improve the manuscript, and once these have all been addressed, I recommend this manuscript to be accepted for publication.*

*General Suggestions*

*In the introduction, where the chemical processes are explained, the writing style can feel at times a bit like a textbook. It would be good if the authors could show their awareness of the existing body of literature by citing a handful of papers which discuss the individual aviation-NOx related chemical processes.*

*When the methodology is explained it would help if the authors could add more information about the rationale behind the choices they have made for the design of their study. My specific comments below highlight the relevant parts in the manuscript. While many papers will build on work from previous publications (all are duly referenced in this manuscript, however many of them are often linked to the same affiliation) the methods need to be sufficiently explained in a stand-alone way, so that the reader (or the reviewer) do not need to consult a chain of previously published papers in order to be able to follow.*

*Some of the methods used in this study appear quite novel and the application of Machine Learning (ML) techniques is becoming increasingly prevalent. Given the novelty it would be good if the authors could again help the reader understand why the use of ML techniques brings advantages to this particular study.*

*The impact of the applied NOx perturbations, released at aircraft cruise altitudes, will be highly dependent on the modelled background atmosphere, especially the NOx mixing ratios. The abilities of global models to simulate the NOx background accurately still remains a challenge. Further, the fact that the meteorology of only one year has been used for this study can provide a limitation. This latter point is not a criticism (given the computational expense of the experiments) however it would be good if the authors could acknowledge these limitations in the discussion and in the conclusions.*

**We appreciate the constructive feedback provided by the reviewer as it has helped us improve the manuscript. Overall, we have condensed the section in which extensive detail regarding the chemical reactions that are relevant for aviation $NO_x$ in the troposphere was given, and expanded the literature covered. Additional information regarding the climate-chemistry model that was used and the machine learning methods have been included. Further discussion on the limitations of our research is also provided. Please find our replies for each topic below.**

*Specific comments*

*Lines 40—54:* This paragraph would benefit from more citations that will provide more details on the individual mechanisms through which aviation emissions affect the climate system

**We have reworded a part of the introduction to only provide an overview of the relevant processes and have expanded the literature included in this section for the individual mechanisms through which aviation emissions affect the climate system (lines 40 – 57).**

*Line 48: Does aviation NOx release into the stratosphere result everywhere (in the stratosphere) in catalytic O3 loss? My understanding is that this is particularly relevant at altitudes above the lower stratosphere (and above typical subsonic aircraft cruise altitudes). For emissions into the LS I would expect O3 production to occur. Again, a citation here would be helpful that explores what happens to aviation NOx released into the stratosphere.*

**It is indeed worth specifying that $NO_x$ will not lead to $O_3$ destruction everywhere within the stratosphere. Typically, within the upper troposphere/lower stratosphere (UT/LS), the lower $NO_x$ background concentrations lead to a quasi-linear $O_3$ production process (Matthes et al., 2022)[5]. The $O_3$-neutral altitude can be found slightly above the UT/LS, at a range of about 13 – 14 km (with a global mean ozone-neutral altitude of 13.5 km), in which emitted $NO_x$ would lead to a net null $O_3$**

[5]Matthes, S., Lee, D.S., De Leon, R.R., Lim, L., Owen, B.; Skowron, A., Thor, R.N., Terrenoire, E.: Review: The Effects of Supersonic Aviation on Ozone and Climate. Aerospace 2022, 9, 41. https://doi.org/10.3390/aerospace9010041.

**disturbance as its tropospheric production is counteracted by its stratospheric depletion. Figure 1 from Fritz et al. (2022)[6] illustrates this point. This is now clarified in lines 50 – 54.**

*Line 56: Both ERF and RF are used in this paragraph but not well explained. The authors should either explain the difference between these concepts or cite a publication where the reader can learn about it.*

**We have added a citation to the 2013 IPCC report[7], which includes these definitions (line 59).**

*Line 64: "Here…" I believe the authors are referring to Stevenson & Derwent (2009) so this should be "Therein…"?*

**This change has been incorporated.**

*Line 80: Just to clarify: this sentence seems to imply that O3 reaches only at the end of its lifetime (after 3 months) its location of largest impact. Are the authors suggesting that there is less radiative impact prior to this time?*

**We did not mean to say that the main location of impact occurs exactly after 3 months, in actuality, the $O_3$ peak (as can be seen from Fig. 15 (e)) tends to occur approximately after 15 days from emission for most cases. We have therefore rephrased the original sentence to clarify (now in lines 84 - 86):**

**"In summary, none of these examples have comprehensively studied the complete and intercontinental journey of aviation $NO_x$ from its point of release, intermediate transport pathways, until most of the resulting $O_3$ is removed from the atmosphere 3 months later (typical $O_3$ lifetime), and the associated climate forcing."**

*Line 88: The way the process is described here seems to assume that the content of the air parcels with their chemical evolution remain completely isolated from the environment, in reality however some kind of mixing would occur.*

**We now mention that the total number of 170 000 Lagrangian air parcels is chosen per simulation specifically to ensure more realistic transport and inter-parcel mixing conditions. 1 400 air parcels are responsible for the initial transport of the emitted $NO_x$ while the remaining 168 600 serve the primary purpose of modelling the mixing that occurs between air parcels. We have clarified this in lines 93 – 95. This is also mentioned in Section 2.1 in lines 172 – 173.**

**Further information about this process is provided in Section 2.1, in which the sub-model responsible for inter-parcel mixing, LGTMIX, is described and a reference (Brinkop and Jöckel, 2019)[8] is also provided (lines 175 – 177).**
* * *
[6]Fritz, T.M., Dedoussi, T.C., Eastham, S.D., Speth, R.L., Henze, D.K., and Barrett, S.R.H.: Identifying the ozone-neutral aircraft cruise altitude. Atmospheric Environment, Vol. 276, https://doi.org/10.1016/j.atmosenv.2022.119057, 2022.

[7]IPCC, 2013: Climate Change 2013: the Physical Science Basis, Contribution of Working Group I to the Fifth Assessment Report of the Intergovernmental Panel on Climate Change. Stocker, T.F., Qin, D., Plattner, G.-K., Tignor, M., Allen, S.K., Boschung, J., Nauels, A., Xia, Y., Bex, V., Midgley, P.M.: Cambridge University Press, Cambridge, United Kingdom and New York, NY, USA, 2013.

[8]Brinkop, S. and Jöckel, P.: ATTILA 4.0: Lagrangian advective and convective transport of passive tracers within the ECHAM5/MESSy (2.53.0) chemistry-climate model, Geosci. Model Dev., 12, 1991-2008, https://doi.org/10.5194/gmd-12-1991-2019, 2019.

*Line 89: This confused me a bit. Surely, it is not the Lagrangian methodology per se that results in terabytes of data but the selected number of trajectories/parcels and, in connection with that, the number of simulated processes?*

**Although the Lagrangian methodology itself is not the direct cause for the large amounts of data, it does demand close to 170 000 air parcel trajectories in order to realistically model the transport from the emission points (1400 air parcels) and the mixing between parcels (~169 000 air parcels). All of these trajectories will then have their own 4D fields (function of time, altitude, latitude and longitude) for multiple chemical species (e.g. $O_3$, OH, $HO_2$, $CH_4$, …) that will quickly add up to several terabytes of data overall.**

**We have rephrased this sentence to make this clearer:**

**"Such an approach, however, naturally yields large amounts of data (in the order of terabytes) from the necessity to include close to 170 000 air parcels per simulation to ensure realistic transport and inter-parcel mixing. All of these trajectories can, however, be efficiently processed in practice with the integration of unsupervised machine learning techniques such as clustering."**

*Line 90: Could the authors add some text explaining why they think ML techniques like clustering offer an advantage to process the data? I am not disputing this, but it will not seem obvious to all readers. Why will ML techniques be helpful for this?*

**Machine learning techniques like clustering offer a systematic way of identifying patterns within very large datasets. In this context, that means being able to meaningfully group together the thousands of trajectories transporting the emitted $NO_x$ into different transport pathways. As a simple example for emissions in North America, without any clustering algorithms, it would be difficult to categorize the air parcels as can be seen in the figure below.**

[Figure]

**Figure R1: Air parcel trajectories plotted in blue for emissions in North America**

**A sentence has been added to explain this (lines 95 – 98).**

**"All of these trajectories can, however, be efficiently grouped together with the integration of unsupervised machine learning techniques such as clustering. In other words, it offers a systematic way of categorizing thousands of air parcels based on common features, which ultimately allows for a faster identification of patterns in very large datasets."**

*Line 138: How were the chemical fields for these 10 simulations initialised? Were the initial conditions for these simulations specific to 2014?*

**The various background chemical fields are taken from activities of Phase One of the chemistry-climate model initiative (CCMI-1) data for the year of 2014. For more realistic atmospheric conditions, our simulations were run using nudging (Newtonian relaxation) of the vorticity, the wind divergence, the logarithm of the pressure field at the surface and the temperature towards 2014 ERA-interim reanalysis data. Further details explaining the variables that were nudged has been added to Section 2.1 (lines 146 – 148).**

*Line 144 & 145: It would be good if the authors could explain some of their choices for timing and magnitude of the NOx perturbations. Why 0600 UTC? Why 5x10^5 kg NO, how does this relate to average aircraft NOx emissions released over, say, the United States and what is a typical NOx background? Is this a doubling of emissions or does it represent a massive spike? That probably depends on the location. It would be good to have some context or explanation where this number comes from.*

**While any emission time could have been chosen, the choice for the emission time of 0600 allows us to more directly compare with previous studies that have also used the same starting point. We have added a sentence in Section 2.1 as clarification (lines 154 – 156).**

**To contextualize the amount of NO (0.5 Gg) that we emit at each emission point shown in Fig.1, we have included below a graphical comparison with total yearly NO emissions at 250 hPa from commercial aviation during the period 2017 – 2020 according to the most recently available aviation emissions inventory[9]. A few sentences explaining this contextualization have also been incorporated in Section 2.1 (lines 159 – 163).**

**"This amount of emitted NO may be compared to total yearly NO emissions at cruise (~250 hPa) by commercial aircraft for all five regions (defined in Fig. 1) to the most recently available aviation emissions inventory (Quadros et al., 2022b). According to this inventory, the emitted NO amount of 0.5 Gg constitutes roughly 40% of mean total yearly emissions by commercial aircraft in N. America, 32% for Eurasia, 186% for S. America, 323% for Africa and 118% for Australasia."**
* * *
[9]Quadros, F.D.A., Snellen, M., Sun, J. and Dedoussi, I.C.: Global civil aviation emissions estimates for 2017-2020 using ADS-B data. Journal of Aircraft, https://doi.org/10.2514/1.C036763, 2022b.

[Figure]

**Figure R2: Total yearly commercial aviation NO emissions at an altitude of about 250 hPa by region as defined/shown in Fig. 1. NO₂ data has been obtained from Quadros et al. (2022b) and converted into equivalent NO mass. The value for 2017 has been scaled by a factor 2 as the emissions were limited to the period July 2017 – December 2017. The red dotted line represents the constant amount of NO ($5\times10^5$ kg) that we emit in this study at each emission point.**

We do highlight, however, that for the linearized sub-model AIRTRAC that we use to calculate the contributions to the atmospheric concentration of $O_3$ by $NO_x$, the amount itself is not relevant as the output will be scaled linearly (lines 163 – 164).

A comment on what typical background $NO_x$ levels might be in our model at 250 hPa is included in Section 2.1 (lines 164 – 166):

"In terms of the background $NO_x$ levels, typical volume mixing ratios near N. America in July 2014 at 250 hPa range between $5\times10^{-10}$ and $9\times10^{-10}$ mol·mol$^{-1}$."

*Figure 1:*

*Obviously the authors chose a more conceptual approach by shaping their regions in a way that roughly covers a continent (without overlap) without exhibiting any preferential treatment. This will not reflect actual distributions of aircraft emissions such as the North Atlantic Flight Corridor, major aviation hubs in East Asian cities or any flights across the Pacific. Shaping the regions to match more realistically the existing flight routing would have been an alternative to the chosen approach. Given the focus of this manuscript on assessing aviation climate impacts, as opposed to focusing conceptually on atmospheric processes (independent of the aviation application), is there a specific reason why the authors chose this particular distribution?*

We chose this distribution to have more information on points that are typically not as frequently analyzed for current aviation emissions, which could become important in the near future given the growing nature of the aviation industry in the near term. Predictions of changes in regional flight distributions in terms of revenue passenger kilometers (RPK) estimate decreases from 26% to 17% and from 23% to 17% for North America and Europe, respectively, between the years 2018 and 2050 (Fig. 2 in Gössling and Humpe, 2020[10]). Research is already available for the main regions with
* * *
[10]Gössling, S. and Humpe, A.: The global scale, distribution and growth of aviation: Implications for climate change. *Glob. Environ. Chang.*, https://doi.org/10.1016/j.gloenvcha.2020.102194, 2020.

**current elevated flight traffic like the Northern Trans-Atlantic flight corridor or Europe (Grewe et al., 2014[11]; Frömming et al., 2021[12]; Rosanka et al., 2020[13]; Grobler et al., 2019[14]). By combining our research with such studies, a more comprehensive understanding of aviation climate effects that could already help account for the aforementioned shifting trends in global flight distributions is provided. This argumentation has been added to Section 2.2 to further justify the choice of the emission points in Figure 1 (lines 210 – 217).**

*Section 2.3: The methodology needs a bit more explanation. Radiative forcing from O3 is most effective at the tropopause region (see papers by Shine et al), therefore it is not obvious how linking emission region with region of largest air quality impact (at the ground) relates back to maximum RF impact*

**We would like to clarify that we are not attempting to link the instantaneous radiative forcing calculated at the tropopause with any air quality effects. We understand that this confusion comes from referring the study by Quadros et al. (2020) in which aviation air quality effects are studied via a source-receptor analysis. By including this citation, we hoped to highlight the similarity in methodologies only. We have clarified this point by now explicitly stating that we consider only aviation climate effects and not air quality (lines 221 – 227).**

**"Past studies, within the context of aviation air quality effects, have adopted a similar approach to comprehend if, in terms of aviation-induced $O_3$ and fine particulate matter ($PM_{2.5}$), the source of the emission directly corresponded to the most affected region (Quadros et al., 2020) … Here, we look at aviation climate effects by considering the instantaneous RF is calculated with the RAD sub-model with respect to the climatological tropopause and subsequently averaged over the regions in Fig. 2."**

*Figure 2: The dark green lines and labels over grey background are difficult to read. Can you make them more visible (e.g. in red)?*

**The suggested changes have been made to Figure 2.**

*Line 203: "To identify characteristic patterns" is too brief an explanation and should be expanded into more detail. For readers who are not familiar with clustering it would be good if a bit more text (1-2 sentences) could be added that explains the benefit.*

**We have clarified what we meant by "characteristic patterns" and have explained the benefit of clustering in this particular context (lines 232 – 235):**

**"To systematically group the output trajectories of the Lagrangian approach within EMAC (Section 2.1) in terms of their geometric similarities in altitude and latitude as well as their radiative forcing**
* * *
[11]Grewe, V., Frömming, C., Matthes, S., Brinkop, S., Ponater, M., Dietmüller, S., Jöckel, P., Garny, H., Tsati, E., Dahlmann, K., Søvde, O. A., Fuglestvedt, J., Berntsen, T. K., Shine, K. P., Irvine, E. A., Champougny, T., and Hullah, P.: Aircraft routing with minimal climate impact: the REACT4C climate cost function modelling approach (V1.0), Geosci. Model Dev., 7, 175–201, https://doi.org/10.5194/gmd-7-175-2014, 2014.

[12]Frömming, C., Grewe, V., Brinkop, S., Jöckel, P., Haslerud, A. S., Rosanka, S., van Manen, J., and Matthes, S.: Influence of weather situation on non-CO2 aviation climate effects: the REACT4C climate change functions, Atmos. Chem. Phys., 21, 9151–9172, https://doi.org/10.5194/acp-21-9151-2021, 2021.

[13]Rosanka, S., Frömming, C., and Grewe, V.: The impact of weather patterns and related transport processes on aviation's contribution to ozone and methane concentrations from NOx emissions, Atmos. Chem. Phys., 20, 12347–12361, https://doi.org/10.5194/acp-20-12347-2020, 2020.

[14]Grobler, C., Wolfe, P.J., Dasadhikari, K., Dedoussi, I.C., Allroggen, F., Speth, R.L., Eastham, S.D., Agarwal, A., Staples, M.D. and Sabnis, J.: Marginal climate and air quality costs of aviation emissions. Environ. Res. Lett. 14, 114031, 2019.

**effects, we incorporate a clustering algorithm to the methodology. This enables us to first identify the different types of transport geometries across all regions and then to associate an average radiative forcing estimate to each in an orderly fashion."**

*Section 2.4.1: This section is very technical and not within my area of expertise. I could not follow all parts of this description and, given that the target audience for ACP are atmospheric scientists I would recommend a to include a less technical summary of the methodology without formalism. The same applies to section 2.4.2 about the K- or L-method which are not explained and will likely not be general knowledge to the average atmospheric scientist.*

**To add clarity to the clustering methodology, we have added a paragraph at the end of Section 2.4.1 summarizing some of the main points (lines 286 – 292):**

**"In summary, the QuickBundles algorithm systematically groups median trajectories based on geometric similarities (altitude and latitude) and their climate effects (mean radiative forcing associated with an emission point). The first trajectory simply becomes the first cluster and all subsequent trajectories are allocated based on the calculation of a metric (Eq. 1) that we propose here that accounts for both transport features and climate effects. At every step in the clustering process, the distance between the centroidal trajectory of a cluster and the candidate trajectory is calculated using Eq. 1 and depending on the user-defined parameter $\theta$, it will either be placed within a pre-existing cluster or an entirely new one. The optimal selection of $\theta$ is discussed in the next section."**

**We now also explicitly allude to steps 4 and 5 from Fig. 3 (which were previously not mentioned) in our explanation of the clustering algorithm (lines 281 – 283).**

**In addition, a short description is provided for the K-means clustering method in Section 2.4.2 along with an additional reference (Hartigan and Wong, 1979)[15] (lines 297 – 299).**

**As for Section 2.4.2, additional information regarding the L-method is provided (lines 299 – 311):**

**"The approach chosen here is based on the L-method, developed by Salvador and Chan (2005) and applied by Kassomenos et al. (2010). It consists of finding the intersection between two linear regressions that model each half of the curve generated from plotting the evolution of a trajectory-based metric, specifically here the average distance within clusters, as a function of the cluster number, as is exemplified in Fig. 4 (a). Possible other metrics as described by Kassomenos et al. (2010) could be the SSE, a similarity function, or another distance metric. The red dot in Fig. 4 (a) may be viewed as a partitioning point as it divides the curve in two quasi-linear sections: left and right. The final linear regression (dotted red line) for the points on the left may be obtained iteratively by beginning with the most basic case of regressing the first two, leftmost data points. The fit of such a linear function may then be improved gradually by adding more and more points in each iteration. According to Salvador and Chan (2005), the optimal distribution of points to be regressed by each side could be achieved by iteratively selecting the partitioning point that minimizes the total root mean squared error of both linear regressions. Kassomenos et al. (2010), whose approach we follow here, however, have also tested this method by applying different curve modelling techniques that include higher order polynomials and splines and obtained consistent results relative to the linear approach."**
* * *
[15]Hartigan, J. H. and Wong, M. A.: A K-Means clustering algorithm., Appl. Stat., 28, 100–108, 1979.

*Section 2.5: Can the authors give a little more context why we are interested in the rate of descent?*

**This metric quantifies the average vertical speed of an air parcel trajectory from its point of release at 250 hPa until its minimum altitude. We are interested in this quantity across all regions because it can help us verify if there is a statistically significant correlation between an air parcel's vertical transport and its mean $O_3$ production. We have added a sentence describing the usefulness of this metric in (lines 334 - 336):**

**"The quantity $\dot{H}$ is calculated for all regions in Section 3.1 and is a useful metric when paired with the mean $O_3$ production of air parcels to determine if a faster initial downward transport translates to a smaller impact on atmospheric concentrations of $O_3$."**

*Section 3 – Intro: Can the authors please add an explanation for the connection between air parcel descent and O3 production. Further, why is the latitudinal residence time relevant?*

**We had hypothesized that a faster downward transport of air parcels would be linked to a lower mean $O_3$ production by observing Figs. 6, 7 (a) and 7 (b). A possible explanation for such a phenomenon could be owed to the larger $O_3$ removal rates at lower altitudes compared to the likelihood of a more efficient accumulation of both $NO_x$ and $O_3$ near the emission altitude of 250 hPa. Figure 4 from Grewe et al. (2002)[16] highlights the direct relationship between pressure altitude and $O_3$ lifetime for altitudes below 200 hPa during the months January, April, July and October. A sentence has been added to the introduction of the Results section 3.1 to clarify this point (lines 345 – 348).**

**The distribution of latitudinal residence times (Fig. 8) is also of interest to us as it provides further insight into the resulting transport patterns arising from different global starting points. It shows, for instance, how there is a low tendency for transhemispheric transport and mixing between emissions in the North and the South. The following sentence has been added (lines 424 – 425):**

**"The distribution of latitudinal residence times in Fig. 8 also solidifies the reduced tendency for transhemispheric transport and mixing between air parcels released in the North and in the South."**

*Lines 327: It would be good to have a sentence here providing a transition from the preceding description of Table 1 to the quickbundles results. Otherwise, this reads a bit disjointed, I felt a bit confused when I read this at first.*

**A sentence has been added (lines 383 – 385).**

*Line 342: "area of lower chemical activity" This is a misleading statement. It needs to be clear that this refers to the discussed O3 processes of interest in this study. There is lots of other chemical activity here.*

**This point has been clarified in lines 400 – 401.**

*Figure 9: It is easy to discern the difference in the red. The green coloured trajectories however cover a large area of the plot, and the distinction between the two panels is not so easy to see in the green. Could perhaps the age of the trajectory be expressed through different colour shading, or perhaps some other information be conveyed such that the change in structure becomes more visible?*
* * *
[16]Grewe, V., Dameris, M., Fichter, C., and Lee, D.S.: Impact of aircraft NOx emissions. Part 2: Effects of lowering the flight altitude, Meteorol. Z. 3, 197-205, 2002.

**As suggested, the age of the air parcel trajectories is now more discernible with the variable blue transparency setting. Additionally, the colors used in Figure 9, as well as in Figure 5 in the manuscript and Figures S1-2, S7-8, S13-14, S19-20, S25-26, S31-32, S37-38, S43-44, S49-50, and S55-56 from the supplementary materials have now been changed such that they are more visible (they are also colorblind-friendly).**

*Figures 12 and 13: There is a lot of information in these figures. What springs to mind is that all this information is based on a single year of study. Will the authors consider the possible impact of inter-annual variability, e.g. in circulation patterns. Obviously, without carrying out a multi-year study this would be difficult to quantify. However it would be worth stressing to the reader that the findings are based on the meteorology from a specific year.*

**We have emphasized in the captions of both Figs. 12 and 13 that our findings correspond to the meteorological conditions from two distinct time periods from 2014: January 1 – March 31 and July 1 – September 30. This is also now addressed in the discussion of the limitations of this work (lines 748 – 749) and in the conclusion (lines 770 – 771).**

*Line 524: Here is a statement that puts the NOx perturbation in a practical context. This should have been mentioned much earlier in the discussion of the methodology. "121 700 A320 aircraft", does this mean 121,700 (one hundred and twenty one thousand and seven hundred)? The context is certainly very specific (even mentioning to a specific engine model) but what does this mean in terms of percentage increase of contemporary air traffic? Please do not misunderstand me, I welcome the fact that the authors try to put it into context, however adding a more easily accessible context (to the non-expert) would be appreciated, something along the lines of "this corresponds to a 200% increase in emissions from transatlantic flights from mid-sized airliners" or something similar.*

**We agree that this sentence is too specific and is therefore not as helpful as we had hoped initially. Therefore, we have removed it and instead have briefly reiterated the context given in Section 2.1 (lines 604 – 606) about how an emission of 0.5 Gg of NO compares to total yearly emissions by commercial aircraft at 250 hPa:**

**"As a comparison, this $NO_x$ perturbation in each emission point represents about 40% of the mean total yearly NO emitted by commercial aviation in N. America (as defined by the boundaries in Fig.1) between the years 2017 – 2020 at a pressure altitude of 250 hPa […]"**

**Please see our reply to the earlier comment on (original) lines 144 & 145 for further information (p. 9-10 of this document).**

*Line 534: Careful with general statements: this is based on NOx-O3 processes only, ignoring other aviation related non-CO2 processes. If this is not mentioned this statement could be misunderstood.*

**We have rephrased this sentence so that it now becomes clear that we are referring specifically to only the short-term increase in $O_3$ from emitted $NO_x$ (lines 612 – 615).**

*Line 555: Could the authors also add a bit of interpretation for the link between N. America as a source region and Europe as impact region. Subsequently an interpretation is offered for the southern hemisphere, however it would be also good to have some text about this largest impact.*

**We have expanded on our interpretation regarding Europe bearing the largest mean RF impact from an emission in N. America. In summary, several factors are involved: local emission time, the meteorology, the dimensions of the receptor regions and also the background atmospheric**

composition along the trajectories as this will impact the $O_3$ productive efficiencies. As the RF impact is directly linked to the amount of $O_3$ produced per receptor region, we have calculated the total amount of time spent by $NO_x$-carrying air parcels specifically during their maximum $O_3$ production (occurring between 14.25 – 20 days on average for emissions in N. America during July) averaged by the area of each receptor region. This is summarized in the newly added Table 3. Overall, it is therefore likely that Europe experiences the largest RF impact since these NOx-carrying air parcels spend the most time during their peak $O_3$ production per unit area within Europe ($5.52\times10^{-4}$ hr·km$^{-2}$). We first discuss this in lines 636 – 643 and again in the Discussion section (lines 710 – 713).

*Section 4: The first part of this section focuses mostly on the experience gained from the methodology which is a good idea. The second part provides a fairly high-level discussion of the scientific findings. Given the complexity of the study design and the employed methodology, this work has yielded in a large number of results. While Section 3 has largely been confined to presenting the quantitative findings, not much space was dedicated to an interpretation. I would have hoped to find more of this in Section 4. If the authors do not wish to expand Section 4 then I would recommend that within Section 3 a few more sentences of interpretation are added to specific findings. In general, I am satisfied that the authors provide a discussion of the caveats and potential limitations of their work.*

Both Sections 3 and 4 have now been complemented with additional analysis of our findings. In terms of Section 3, we have expanded our interpretation of the global clustering results for the horizontal transport distributions during both time periods during January (Fig. 12) and July 2014 (Fig. 13) in terms of the atmospheric circulation patterns shown in Appendix C (lines 542 – 552 for Section 3.2.1 and lines 585 – 591 for Section 3.2.2). In terms of Section 4, we have also stated that our clustered results are strongly influenced by the dominant westerlies in both the Northern and Southern Hemispheres. While extreme weather events with intensified wind shear are likely to bias our results, the El Niño Southern-Oscillation (ENSO), however, is unlikely to have acted as a bias in 2014 (lines 689 – 694).

*An important point is that the diagnosed impacts of a NOx perturbation will be strongly dependent on the simulated NOx background in the chemistry-climate model. While a general validation of the model's atmospheric background is beyond the scope of this work this needs to be mentioned. It would be ideal if the authors could point to a model validation paper which gives a fair indication of the performance of their model in this regard.*

We have included three additional references that validate the chemistry setup for our model, which includes an evaluation of the $NO_x$ tropospheric tracer (lines 724 – 729).

*Line 620: To what extent do the authors expect the linearisation to hold true for their perturbation? Are we well within range of a linear response for small perturbations? Has this been explored, if not by the authors then by other studies perhaps?*

The AIRTRAC sub-model[17,18] referred within the manuscript calculates the contributions to atmospheric concentrations of several species like $O_3$ in a linearized manner. We have performed an additional simulation where we perturb emissions by 10% of the original amount ($5\times10^5$ kg (NO)) to
* * *
[17]Supplement from Grewe et al., 2014

[18]Grewe, V., Frömming, C., Matthes, S., Brinkop, S., Ponater, M., Dietmüller, S., Jöckel, P., Garny, H., Tsati, E., Dahlmann, K., Søvde, O. A., Fuglestvedt, J., Berntsen, T. K., Shine, K. P., Irvine, E. A., Champougny, T., and Hullah, P.: Aircraft routing with minimal climate impact: the REACT4C climate cost function modelling approach (V1.0), Geosci. Model Dev., 7, 175–201, https://doi.org/10.5194/gmd-7-175-2014, 2014.

**showcase this. The figures below help to show how scaling the $NO_x$ emission by a certain factor will also scale the $O_3$ production in the same way.**

[Figure]

[Figure]

**Figure R3(a) Two $NO_x$ emission scenarios**

**Figure R3(b) $O_3$ response to the two $NO_x$ emission scenarios**

**Figure (a) shows two different NOx emission scenarios in which 100% of the emissions ($5\times10^5$ kg(NO); blue curve) are applied and then only 10% ($5\times10^4$ kg(NO); green curve) are released at 300 hPa. Figure (b) then shows the corresponding $O_3$ production curves to each of these scenarios. As is shown by the orange line, the ratio between each point from both $O_3$ curves is constant and equal to 10. This is precisely the value we would expect in a linearized response since $\frac{5\times10^5}{5\times10^4} = 10$. We believe that the linearization of AIRTRAC is applicable given that we are emitting $NO_x$ in an altitude (UT/LS) in which the short-term $O_3$ production is quasi-linear[1]. AIRTRAC itself has therefore previously been applied to analyze the climate impact of $NO_x$ and $H_2O$ emissions from similar altitudes[5,19]. This is now also clarified in the text (lines 163 – 164).**

*Line 627: "Earlier work has shown..." It is not clear what the authors mean without reading e.g. the Grewe et al 2019 paper. Can the authors add a bit more explanation?*

**We have added more information regarding the perturbation approach and the source contribution method along with a reference where the interested reader can gain a more in-depth understanding (lines 732 – 741):**

**"We also note that large differences may result depending on the method (either contribution or perturbation) chosen to assess aviation's climate effects. A contribution or source apportionment method consists of finding the amount contributed by a certain emission source to the concentration of a chemical species while a perturbation analysis involves estimating the impact of an emission source on the concentration of a chemical species using two simulations: the first contains all emissions (including, for instance, aviation) and the second either switches off this additional emission or reduces its amount. In a linear scenario between emission and resulting concentration, both methods are theoretically equivalent. However, within the highly non-linear context of $NO_x$-$O_3$ chemistry, significant differences are expected (Clappier et al., 2017). Earlier work has shown that if, for instance, the latter method is used to study the contribution of a sector on atmospheric ozone levels, an underestimation by a factor of up to 7 may result for aviation (Grewe et al., 2019) and up to 2.8 for near-surface sectors (Dedoussi et al., 2020)."**

[19]Frömming, C., Grewe, V., Brinkop, S., Jöckel, P., Haslerud, A. S., Rosanka, S., van Manen, J., and Matthes, S.: Influence of weather situation on non-CO2 aviation climate effects: the REACT4C climate change functions, Atmos. Chem. Phys., 21, 9151–9172, https://doi.org/10.5194/acp-21-9151-2021, 2021.

*Line 645: If more computational resources were available, would the authors prioritise vertical level increases over horizontal grid resolution?*

**Given the strong altitudinal dependence between aviation non-$CO_2$ emissions[20] (especially of $NO_x$ as it could induce $O_3$ production, no $O_3$ at all at the $O_3$-neutral altitudes or $O_3$ destruction in the mid to upper stratospheric region) and their climate impact, we would prioritize an increase in vertical resolution (lines 754 – 757).**

*Line 655: It should be pointed out that the study was based on a single year of meteorology.*

**We have emphasized this in Section 4 (lines 748 – 749) and in the conclusion (lines 770 – 771).**
* * *
[20]Matthes, S., Lim, L., Burkhardt, U., Dahlmann, K., Dietmüller, S., Grewe, V., Haslerud, A.S., Hendricks, J., Owen, B., Pitari, G., Righi, M. and Skowron, A.: Mitigation of Non-CO2 Aviation's Climate Impact by Changing Cruise Altitudes. Aerospace, 8, 36. https://doi.org/10.3390/aerospace8020036, 2021.